# Nymphal diets boost adults' immunity via strengthened constitutive immunity and metabolic capacity in *Adelphocoris suturalis*
Wanning Li, Jingfan Ge, Chong Li, Yuxiao Liu, Letian Xu ✉ & Jing Luo ✉

In insects, juvenile diet strongly influences adult fitness. Yet, despite the centrality of immune function to overall organismal health, the effects of early-life diets on adult immunity remain underexplored. We showed that *Adelphocoris suturalis* adults, developed from nymphs fed on an omnivorous diet (NO), exhibited higher survival against *Beauveria bassiana* than those from a phytophagous diet (NP). Using high-throughput RNA sequencing, we profiled gene expression responses at both early and late infection stages. Integrated transcriptomic and immunological analyses revealed that NO adults exhibited a greater number of up-regulated immune-related genes, elevated phenoloxidase and prophenoloxidase activity, and higher hemocyte counts, particularly during early infection. Tracking macronutrient dynamics and associated gene expression showed that the NO group displayed up-regulation of more metabolic-related genes, which positively correlated with immune activity. These findings highlight a developmentally coordinated link between nutrition and immunity, contributing to an integrated understanding of nutritional and developmental immunology.

The nutritional environments experienced by an animal during its development can have a lasting impact on its phenotype, influencing both immediate and future traits[1,2]. This phenomenon is particularly pronounced in insects, where the juvenile stage is a particularly vital period for nutrient acquisition[3,4]. The nutrients and energy acquired during this stage not only support essential biosynthetic and structural processes in larvae/nymph but also allocated to storage for adequate development and growth in the subsequent non-feeding and adult stages[5,6]. This implies that the nutritional environments during juvenile stage can impact fitness-related traits well into adulthood. For example, in *Drosophila melanogaster*, larvae developed on low protein diets have lower adult weight, smaller adult wing area, and femur size[7]. Similarly, more extended longevity and higher female fertility in *D. melanogaster* adults have been observed by Savola et al., when using an intermediate protein-to-carbohydrate (P:C) food as the larval diet[8]. However, the life-history traits above typically explain only a portion of the fitness. Immunity is also thought to be a core component of fitness in organisms as the ability to combat infection and disease increases survival[9]. Although substantial evidence in insect has shown that dietary nutrients can affect individual immunity[10,11], the potential effect of juvenile diet on adult immunity is rarely considered.

Insects lack adaptive immunity and instead rely primarily on two major forms of innate immunity—constitutive and induced—to combat pathogens. Constitutive immunity operates continuously and provides baseline protection even in the absence of infection, primarily through circulating hemocytes and the basal activity of the prophenoloxidase (PPO) cascade. In contrast, induced immunity is activated upon infection, typically amplifying existing constitutive factors and triggering the production of effector molecules such as antimicrobial peptides (AMPs) and lysozymes[12–14]. Specifically, pathogen-associated molecular patterns are detected by pattern recognition receptors (PRRs), including C-type lectins (CTLs), peptidoglycan recognition proteins (PGRPs), and Gram-negative binding proteins (GNBPs). This recognition initiates a cascade of serine proteases that converts PPO to active phenoloxidase (PO), leading to melanization and pathogen sequestration[15]. Hemocytes proliferate, differentiate, and are mobilized to sites of infection to mediate phagocytosis, nodulation, and encapsulation, either via opsonin-dependent mechanisms (e.g., CTL-mediated recruitment)[16] or direct pathogen recognition[17]. In parallel, pathogen-specific signaling pathways are activated. For instance, fungal infections activate the Toll pathway via proteolytic cleavage of Spätzle (Spz), leading to downstream signaling

State Key Laboratory of Biocatalysis and Enzyme Engineering, School of Life Sciences, Hubei University, Wuhan, China. ✉e-mail: letian0926@163.com; luojing@hubu.edu.cn

through the MyD88-Tube-Pelle cascade and induction of AMPs and lysozymes[18,19].

*Adelphocoris suturalis* (Hemiptera: Miridae) is a notorious pest that can cause serious damage to various crops, including cotton, soybean, maize, and fruit tree. In addition to feeding on plants, *A. suturalis* exhibits facultative zoophagy by preying on small insects or insect eggs, a behavior shown to enhance reproductive fitness[20,21]. This omnivorous feeding strategy allows for flexible manipulation of juvenile dietary inputs by combining plant- and animal-derived components with distinct nutrient compositions. Combined with a functionally complete yet experimentally tractable immune system[20,22], these features make *A. suturalis* an ideal model for studying how early-life diet shapes adult immune function in insects.

In this study, we used *A. suturalis* and *Beauveria bassiana* (a pervasively distributed entomopathogenic fungus that infects insects through cuticular penetration by conidia, followed by hyphal proliferation within the hemocoel) to investigate whether and how juvenile diet shapes adult immune function. Nymphs were reared on either an omnivorous diet (NO, a balanced diet consisting of mung bean sprouts and aphids) or a phytophagous diet (NP, a restricted diet with mung bean sprouts only). We first assessed the survival of adults derived from these dietary treatments following fungal infection. The NO group exhibited significantly higher survival than the NP group. We hypothesized that this survival advantage may reflect diet-induced enhancement of immune capacity. To test this, we used RNA-seq to compare gene expression profiles across dietary treatments at early and late infection stages, focusing on how juvenile diet modulates the adult transcriptional response to fungal challenge. We further assessed immune function by quantifying PO and PPO activity, hemocyte counts, and the expression of representative immune genes. Finally, we measured

macronutrient (carbohydrates, lipids, and proteins) content and the expression of associated metabolic genes to examine whether immune performance was linked to dietary regulation of host macronutrient reserves and associated metabolic processes.

## Results

### Macronutrient composition of the dietary ingredients used in nymphal treatments

To characterize the macronutrient context of the nymphal diets, we quantified total protein, carbohydrate, and triglyceride content of the two dietary ingredients provided during nymphal development: mung bean sprouts and pea aphids (*Acyrthosiphon pisum*). The results showed that the total protein and triglyceride content in aphids was significantly higher than that in mung bean sprouts (Fig. 1b, *t*-test, $P = 0.00072$; Fig. 1d, *t*-test, $P = 1.6 \times 10^{-14}$), while the total carbohydrate content was significantly lower (Fig. 1c, *t*-test, $P = 0.00042$).

### Omnivorous diet at nymphal stage enhances adult resistance to *Beauveria bassiana*

To evaluate the impact of nymphal diets on adult resistance to infection, we assessed the survival of *A. suturalis* adults from different nymphal diet treatments following infection with *B. bassiana* (Fig. 1a, b). The results showed that, compared to the mock control (Ctrl, adults treated with Tween 80), the survival rates of adults infected with *B. bassiana* (Bb) were significantly lower in both dietary treatments (Fig. 1d, Log-rank test, $P = 2 \times 10^{-12}$, $P = 0.0002$; Fig. 1f, g). Notably, adults from the NP group with pathogen infection (NP-Bb) exhibited a significantly shorter median lethal time (LT50 = 5.676 ± 0.052 days) compared to those from the NO group

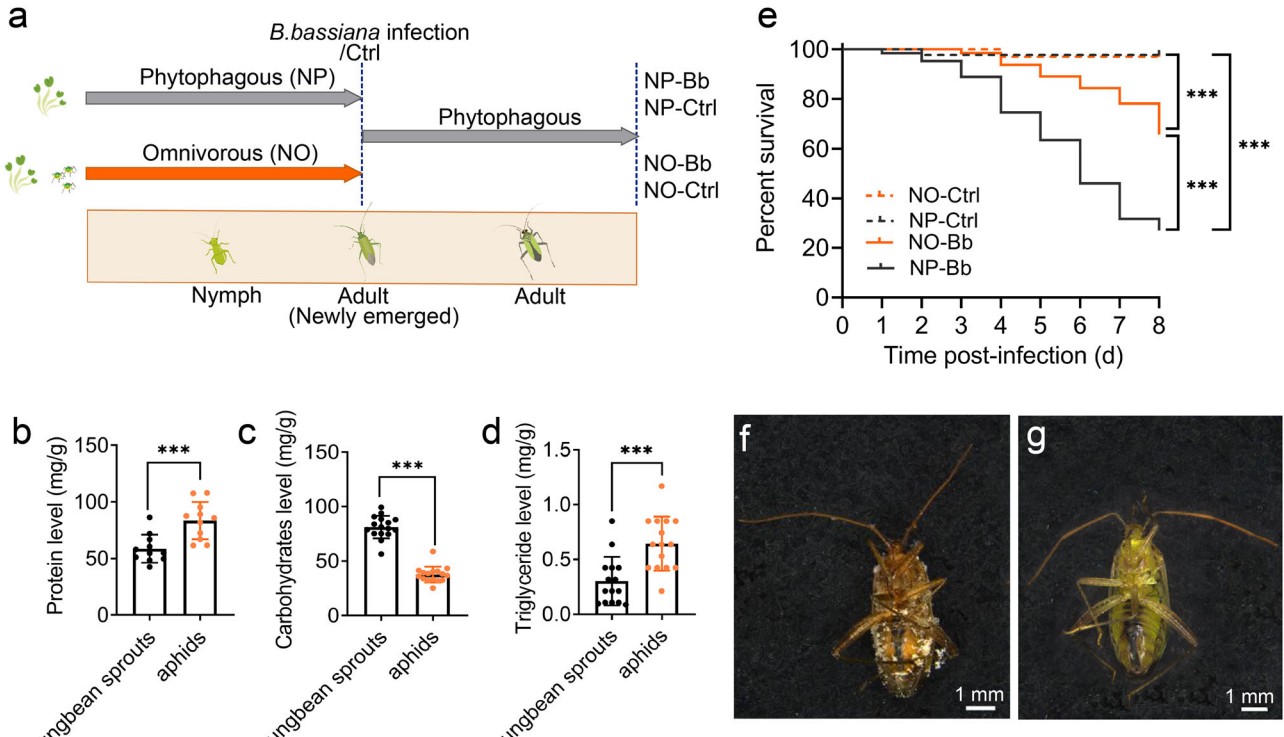

**Fig. 1 | Effects of nymphal diet on the adult resistance to *Beauveria Bassiana* in *Adelphocoris Suturalis*. a** Schematic of the experiment. The nymphs were reared on either an omnivorous diet (NO, a balanced diet consisting of mung bean sprouts and aphids) or a phytophagous diet (NP, a restricted diet with mung bean sprouts only). Newly-emerged adults were either exposed to the *B. bassiana* (Bb) or Tween 80 (Ctrl), and reared on a uniform diet. The total protein (**b**), carbohydrate (**c**), and triglyceride (**d**) content of mung bean sprouts and aphids. **e** Survival of *A. suturalis* adults from different nymphal diets following *B. bassiana* infection. (total protein: *n* = 11 biologically independent samples; carbohydrate: *n* = 16 biologically independent samples; triglyceride: *n* = 15 biologically independent samples; percent survival: *n* = 30–62 biologically independent samples). **f**, **g** Photographs of external symptoms of *A. suturalis* adults infected with *B. bassiana* (left) and Tween 80 (right). Log-rank tests were employed to compare and analyze the survival curves. Student's *t*-test was used to evaluate the significance of differences between two groups (***$p < 0.001$), and data are depicted as means ± SEM.

(NO-Bb), who had a median lethal time exceeding 8 days (Fig. 1d, Log-rank test, $P = 2 \times 10^{-7}$; Supplementary Fig. 2, Log-rank test, $P = 6 \times 10^{-5}$, $P = 9 \times 10^{-5}$). These findings suggest that nymphal diet significantly affects adult resistance to infection, with an omnivorous diet during the nymphal stage substantially enhancing resistance to *B. bassiana* infection.

## RNA-Seq uncovers nymphal diet- and infection-driven gene expression profiles

To investigate the reasons behind how the nymphal stage diet shapes adult resistance to pathogenic fungi, we conducted deep RNA-seq on *A. suturalis* samples from the NO-Bb, NO-Ctrl, NP-Bb, and NP-Ctrl treatments at three time points: 0 days post-infection (dpi), representing the pre-infection baseline; 2 dpi, marking the early infection phase (~10% mortality in NP-Bb group); and 6 dpi, corresponding to the mid-to-late stage of infection progression (~50% mortality in NP-Bb group). A total of 96,293 unigenes were obtained, with the average length of 1,070.11 bp (Supplementary Table 2). Comparing the NO and NP groups revealed 1530 differentially expressed genes (DEGs), with 970 up and 560 downregulated (Fig. 2a). Among them, 512 DEGs were found in the uninfected group (NO-Ctrl vs NP-Ctrl) and 1018 DEGs in the infected group (NO-Bb vs NP-Bb). The greatest number of DEGs, 993, was observed in the infected group at 2 dpi. Volcano plots further indicated that the $-\log_{10}$ [adjusted *p*-value] values were significantly higher in the infected group at this time point, indicating that the DEGs were more significant. These significantly different genes are more likely to participate in the regulatory processes of nymphal diet shaping adult immunity (Supplementary Fig. 1a–e).

The Venn diagram result showed that only nine DEGs were observed in any two time points of uninfected group, and no DEG was consistently observed across all time points in both the uninfected and infected groups (Fig. 2b, d). To gain deeper insights into the functional roles of these DEGs, we performed KEGG pathway analysis separately for the uninfected and infected groups. Result showed that the DEGs of uninfected group were classified into five main categories with 15 subcategories (Fig. 2c), while the DEGs of infected group were classified into four main categories with 13 subcategories (Fig. 2e). Notably, DEGs from both groups were significantly enriched in pathways related to the Endocrine system, Immune system, Digestive system, Amino acid metabolism, Lipid metabolism, Carbohydrate metabolism and Signal transduction. (Fig. 2c, e).

## Adult with an omnivorous diet at the nymphal stage showed enhanced immunity

Blasting a reference set of insect immune-related genes against the *A. suturalis* genome, we identified 695 immune-related unigenes, of which 58 were differentially regulated between the NP and NO groups. These DEGs were categorized into four immune processes: pathogen recognition, extracellular signal modulation, intracellular signal transduction, and immune response effectors. Figure 3a presents the expression patterns of these DEGs visualized through a heatmap. Overall, adults from the NO group exhibited a greater number of upregulated immune genes compared to those from the NP, irrespective of infection status. This upregulation trend was particularly pronounced at 2 dpi, where the pathogen exposure significantly increased the expression of numerous immune-related genes in the NO adults. In contrast, only a limited number of genes were upregulated in the NP adults under the same conditions. Additionally, we noted that both NO and NP adults displayed a substantial number of upregulated immune genes at 0 dpi (Fig. 3a).

Some critical candidate genes spanning from pathogen recognizing to transcription of immune effectors controlling including *GNBP1e*, *CTL2*, *SPZ*, *Toll4b*, *Tube2*, *Pelle1*, *cSP16*, *Lys*, *Alo2a* were selected for RT-qPCR validation (Fig. 3b). At 0 dpi, the transcriptional levels of *SPZ*, *Toll4b*, *Tube2*, *cSP16*, *Pelle1*, *Lys* and *Alo2a* were generally higher in NO group compared to NP, although these differences did not reach statistical significance (Fig. 3b and Supplementary Data 1). As the infection progressed to 2 days, a significant interaction between the nymphal diet and infection was observed on the transcriptional levels of *GNBP1e*, *CTL2*, *Toll4b*, *Tube2*, *cSP16*, and

*Alo2a*. Notably, all these genes reached their highest transcriptional levels in NO-Bb group. Although there was non-significant interaction between infection and diet on the expression of *Pelle2* and *Lys*, the main effects of diet and infection were significant. Specifically, *Lys* transcription was significantly upregulated by both infection and NO diet treatment, while *Pelle2* expression was significantly increased by infection alone (Fig. 3b and Supplementary Data 1). At 6 dpi, a significant interaction between the nymphal diet and infection was detected exclusively for *SPZ*, whose transcriptional level peaked in the NO-Bb group. The transcriptional levels of *GNBP1e* and *Tube2* was primarily driven by infection: *GNBP1e* was significantly upregulated, whereas *Tube2* was downregulated. By comparison, the transcriptional levels of *Toll4b* and *Lys* were mainly influenced by diet, with significantly higher transcript abundance in the NO group than in the NP group. Notably, *cSP16* was independently regulated by both infection and diet, showing infection-induced upregulation and higher expression in the NO group (Fig. 3b and Supplementary Data 1).

To comprehensively assess the changes in adult immune response caused by nymphal diet, we further examined physiological immune parameters, including PPO activity, PO activity, and hemocyte counts. At 0 dpi, newly-emerged adults from the NO group exhibited significantly higher PPO activity and hemocyte counts compared to those from the NP (Fig. 4a, *t*-test, $P < 0.001$; Fig. 4c, *t*-test, $P < 0.05$), while no significant difference was detected in PO activity (Fig. 4b). At 2 dpi, a significant interaction between nymphal diet and infection was observed, affecting all measured physiological immune parameters, with the NO-Bb group showing the highest levels, followed by the NP-Bb (Fig. 4 and Supplementary Data 1). At 6 dpi, the peak of hemocyte counts still observed in the NO-Bb group. PO activity was primarily influenced by diet, with the NO group exhibiting higher activity. In contrast, PPO activity declined significantly in both diet groups compared to their respective controls (Fig. 4 and Supplementary Data 1).

## β-1,3-glucan injection confirms that nymphal diet shapes adult immune activation capacity

To directly test whether nymphal diet modulates adult immune activation capacity independent of other pathogen-induced effects, we performed an immune elicitation assay by injecting β-1,3-glucan—a fungal-specific immune inducer[23]—into 1.5-day-old adults from the NO and NP groups. This timing was selected to align both the developmental stage and immune activation window with those of the *B. bassiana*-infected group, accounting for the 24–48 h typically required for cuticle penetration and systemic immune induction during natural infection[24,25].

β-1,3-glucan injection triggered time-dependent upregulation of all tested immune-related genes, with distinct differences between the NO and NP groups. At 6 h post-injection (hpi), only *GNBP1e*, *cSP16*, and *Lys* were significantly upregulated by β-1,3-glucan in the NO group, while only *Lys* was upregulated in the NP group (Fig. 5a, g, h and Supplementary Data 1). At 12 hpi (developmentally corresponding to 2 dpi in the *B. bassiana* infection experiment), all nine tested genes (*GNBP1e*, *CTL2*, *SPZ*, *Toll4b*, *Tube2*, *Pelle1*, *cSP16*, *Lys*, and *Alo2a*) were significantly upregulated by β-1,3-glucan in both dietary groups, with peak expression consistently observed in the β-1,3-glucan-injected NO group (NO-β-1,3-glucan) (Fig. 5a–i and Supplementary Data 1). By 108 hpi (corresponding to 6 dpi), only a subset of genes (*GNBP1e*, *SPZ*, *cSP16*, and *Lys*) remained significantly upregulated by β-1,3-glucan, and their expression levels continued to be higher in NO than NP (Fig. 5a, c, g, h and Supplementary Data 1). In parallel, β-1,3-glucan also induced diet-dependent differences in physiological immune parameters. At 12 hpi, PPO/PO activities and hemocyte counts peaked in NO-β-1,3-glucan group, followed by β-1,3-glucan-injected NP group (NP-β-1,3-glucan) (Fig. 5j–l and Supplementary Data 1). At 108 hpi, hemocyte counts remained highest in NO-β-1,3-glucan adults, while PPO activity declined significantly in both groups compared to controls (Fig. 5j, l and Supplementary Data 1). These results closely mirrored those observed under *B. bassiana* infection, reinforcing the conclusion that nymphal diet modulates the adult insect's capacity for immune activation.

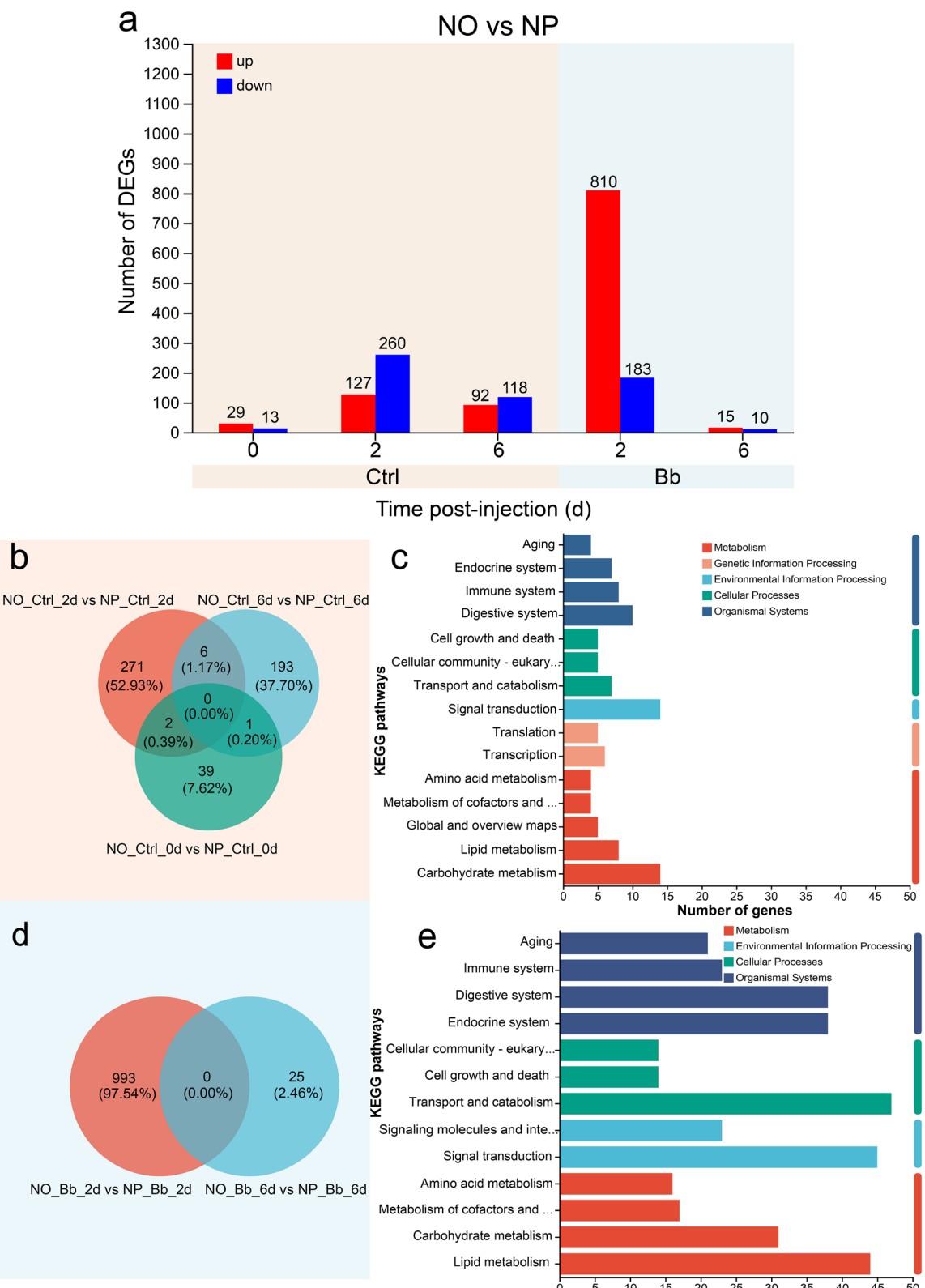

**Fig. 2 | Transcriptome analysis of differentially expressed genes in response to nymphal diet variations. a** Statistical chart of differentially expressed genes. **b**, **d** Venn diagram presenting shared and unique differentially expressed genes at distinct time points under uninfection (**b**) or infection (**d**) conditions. **c**, **e** The KEGG pathway analysis of differentially expressed genes in uninfected (**c**) or infected group (**e**). NO: adults developed from nymphs fed on an omnivorous die; NP: adults developed from nymphs fed on a phytophagous diet; Bb: newly-emerged adults infected with *Beauveria bassiana*; Ctrl: newly-emerged adults treated with Tween 80.

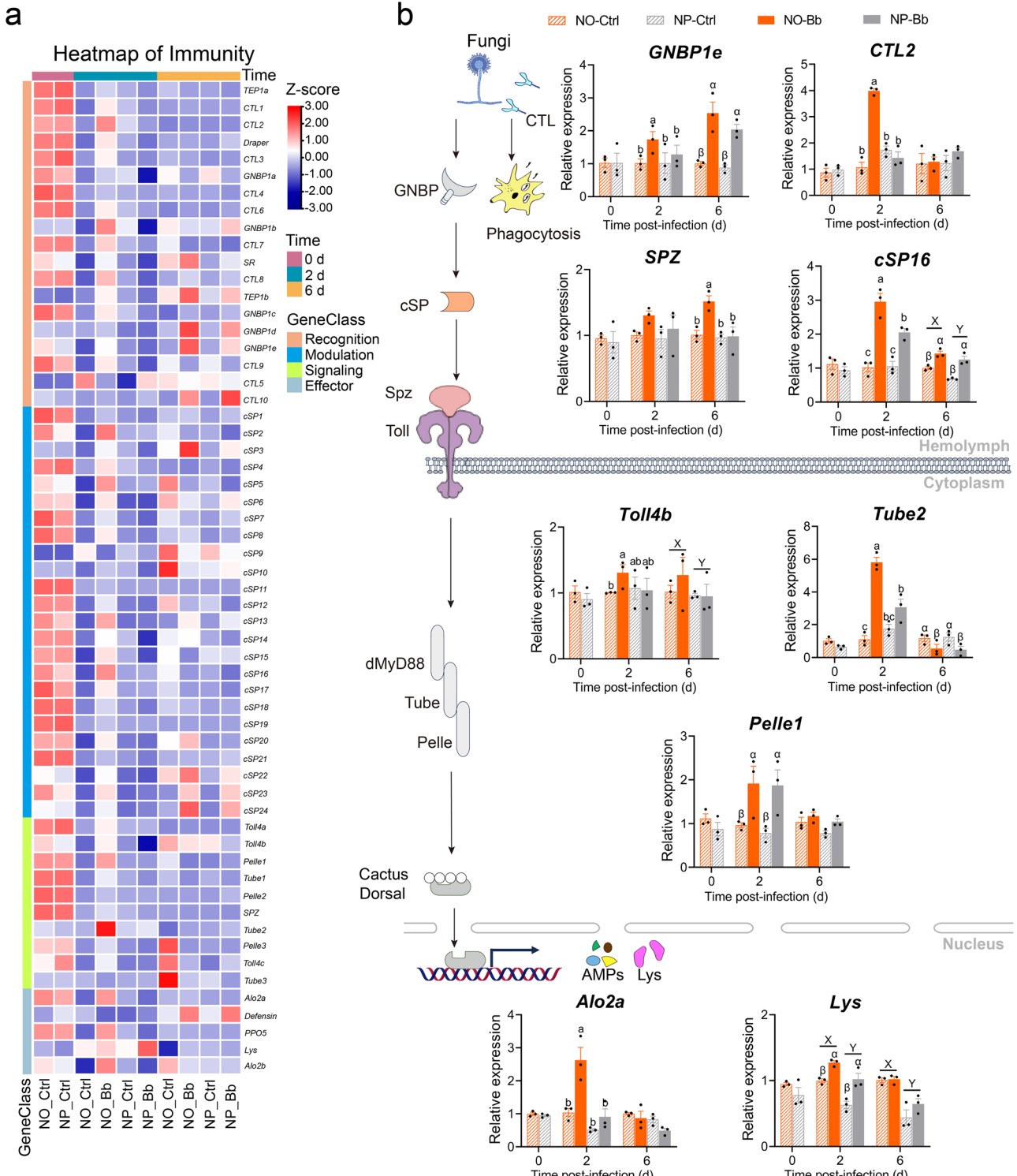

**Fig. 3 | Differentially expressed immune-related genes between NP and NO groups. a** Heatmap illustrating the differential expression of immune-related genes between NP and NO groups at distinct time points. Color scale represents $Z$-scores computed by standardizing the expression of each gene across all samples. Red indicates expression higher than the average level of that gene across samples ($Z > 0$), blue indicates lower expression ($Z < 0$), and white indicates near-average levels ($Z \approx 0$). **b** Validation of representative immune-related gene expression by RT-qPCR. $n = 3$ biologically independent samples. NO: adults developed from nymphs fed on an omnivorous diet; NP: adults developed from nymphs fed on a phytophagous diet; Bb: newly-emerged adults infected with *Beauveria bassiana*, Ctrl: newly-emerged adults treated with Tween 80. Data are presented as means ± SEM. Statistical annotations: different lowercase letters (a–d) indicate significant differences among all treatment combinations when a significant interaction effect was detected; uppercase letters (X, Y) indicate significant differences between diet treatments; Greek letters (α, β) indicate significant differences between infection conditions (Ctrl vs. Bb) ($P < 0.05$).

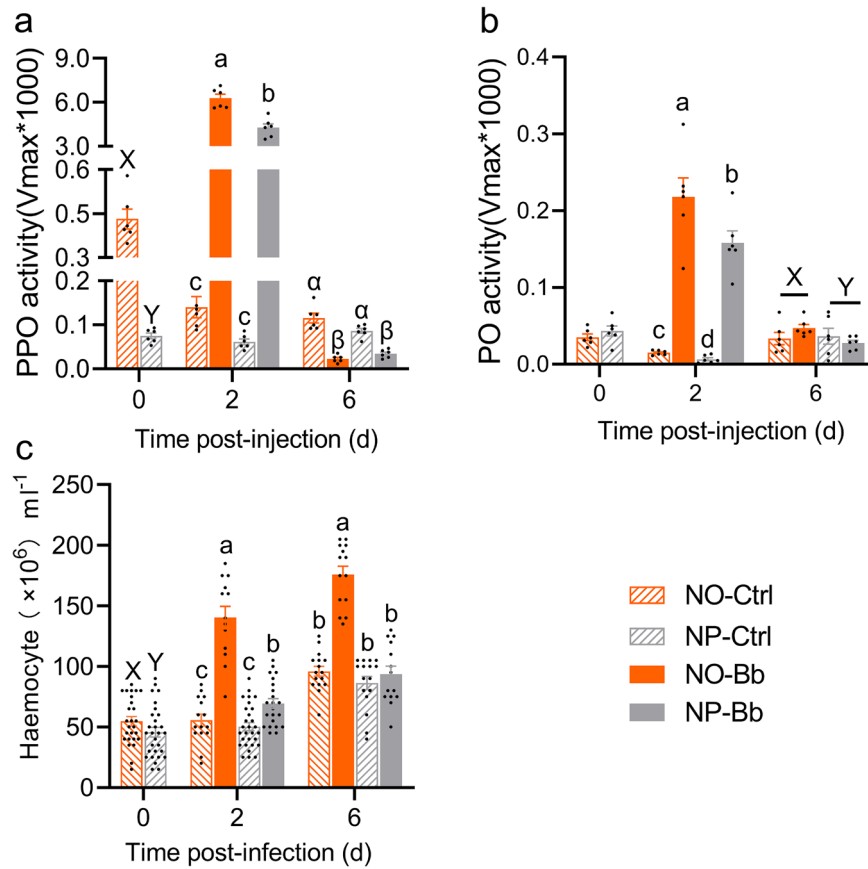

**Fig. 4 | Impact of nymphal diet on adult physiological immune parameters.** Prophenoloxidase (PPO) activity ($n = 6$ biologically independent samples) (**a**), phenoloxidase ($n = 6$ biologically independent samples) (PO) activity (**b**), and hemocyte count ($n = 13–29$ biologically independent samples) (**c**)—were monitored at 0, 2, and 6 days post-infection. NO: adults developed from nymphs fed on an omnivorous diet; NP: adults developed from nymphs fed on a phytophagous diet; Bb: newly-emerged adults infected with *Beauveria bassiana*, Ctrl: newly-emerged adults treated with Tween 80. Data are presented as means ± SEM. Statistical annotations: different lowercase letters (a–d) indicate significant differences among all treatment combinations when a significant interaction effect was detected; uppercase letters (X, Y) indicate significant differences between diet treatments; Greek letters (α, β) indicate significant differences between infection conditions (Ctrl vs. Bb) ($P < 0.05$).

## Metabolic-related genes expression patterns show positive correlation with immune responses

We further monitored the dynamic changes of macronutrient (carbohydrates, lipids, and proteins) content in adults subjected to two nymphal dietary treatments and pathogen exposure, and analyzed the expression patterns of genes associated with the metabolism of these macronutrients.

Nutrient monitoring results revealed that, at 0 dpi, adults from the NO group exhibited significantly higher total protein content (Fig. 6a, *t*-test, $P < 0.001$) and lower triglyceride levels (Fig. 6b, *t*-test, $P < 0.05$) compared to the NP, with no significant differences observed in total carbohydrate content (Fig. 6c, *t*-test, $P = 0.588$) or dry weight (Fig. 6d, *t*-test, $P = 0.852$). By 2 dpi, infection led to a pronounced decline in total protein, triglyceride, and carbohydrate content as well as dry weight across both dietary treatments (Fig. 6 and Supplementary Data 1). It is noteworthy that total carbohydrate content in the NP-Bb group was significantly lower than in the NO-Bb, whereas NP-Ctrl and NO-Ctrl exhibited comparable carbohydrate levels (Fig. 6c and Supplementary Data 1). This suggests that NP adults experience greater carbohydrate depletion under pathogen stress compared to NO adults. In contrast, infection-induced reductions in protein, triglyceride, and dry weight were similar between dietary groups (Fig. 6a, b, d). By 6 dpi, the levels of three macronutrients and dry weight gradually converged across the different treatment groups (Fig. 6 and Supplementary Data 1), which may be due to the consumption of same diet and the end of immune responds.

Heatmap analysis of genes involved in carbohydrate, lipid, and amino acid metabolism revealed parallel expression patterns driven by both nymphal diet and infection (Figs. 7–9a). Across all three metabolic pathways, adults from the NO group consistently showed a higher number of upregulated genes than those from the NP, regardless of infection status. The highest number of upregulated genes was observed in both the NO and NP at 0 dpi, as well as in the NO-Bb at 2 dpi, aligning with the expression trends of immune-related genes. Moreover, at 2 dpi, infection markedly

upregulated numerous nutrient metabolism-related genes compared to uninfected controls. However, by 6 dpi, the expression profile of these genes shifted, with fewer genes showing upregulation in the infected groups compared to earlier time points, suggesting a shift toward a less pronounced metabolic response as the infection gradually subsides (Figs. 7–9a).

Twenty two key genes involved in macronutrient metabolism pathways were selected as candidate genes for RT-qPCR validation, including carbohydrate metabolism-related genes: *Hexokinase 2* (*HK2*), *Glycogen Synthase 59* (*GlyS59*), *Trehalose-6-Phosphate Synthase 1* (*TPS1*), *Phosphofructokinase* (*PFK*), *Glyceraldehyde-3-Phosphate Dehydrogenase 1* (*GAPDH1*), *Pyruvate Kinase 1* (*PK1*), *Isocitrate Dehydrogenase 2* (*IDH2*), *Ketoglutarate Dehydrogenase 3* (*KGDH3*), *Succinate-CoA Ligase 2* (*Scs2*), and *Succinate Dehydrogenase 1* (*SDH1*); lipid metabolism-related genes: *Fatty Acid Synthase 1b* (*FAS1b*), *Long-Chain Acyl-CoA Synthetase 3* (*Acsl3*), *Lipase1f*, *Glycerol-3-Phosphate Acyltransferase 4a* (*Gpat4a*), *Hydroxyacyl-CoA Dehydrogenase 1* (*HADH1*); and amino acid metabolism-related genes: *Glutamate Synthase 1d* (*GOGAT1d*), *Alanine Aminotransferase 2* (*ALT2*), *Glutamine Synthetase 2a* (*GS2a*), *Glutamate Cysteine Ligase 3* (*GCL3*), *Phenylalanine Hydroxylase 1* (*PAH1*), *Tyrosine Hydroxylase 1* (*TH1*), and *Procollagen-Lysine, 2-Oxoglutarate 5-Dioxygenase* (*PLOD*).

At 0 dpi, the expression levels of all these nutritional metabolism-related candidate genes, except for *GAPDH1*, were higher in the NO group compared to the NP, with significant differences observed in *GlyS59, TPS1, IDH2*, and *FAS1b* (Figs. 7–9b and Supplementary Data 1). These findings suggest that the nymphal diet influences the basal expression of metabolic genes in adult, with the NO group showing a stronger activation of metabolic pathways even in the absence of pathogen exposure. At 2 dpi, the majority of nutritional metabolism-related candidate genes, including carbohydrate metabolism-related genes (*HK2, TPS1, GAPDH1, PK1, SDH1, IDH2, Scs2, KGDH3*), all lipid metabolism-related genes, and amino acid metabolism-related genes (*GOGAT1d, ALT2, GS2a, PAH1, TH1, PLOD*), exhibited the highest expression levels in the NO-Bb. In the NO group, most

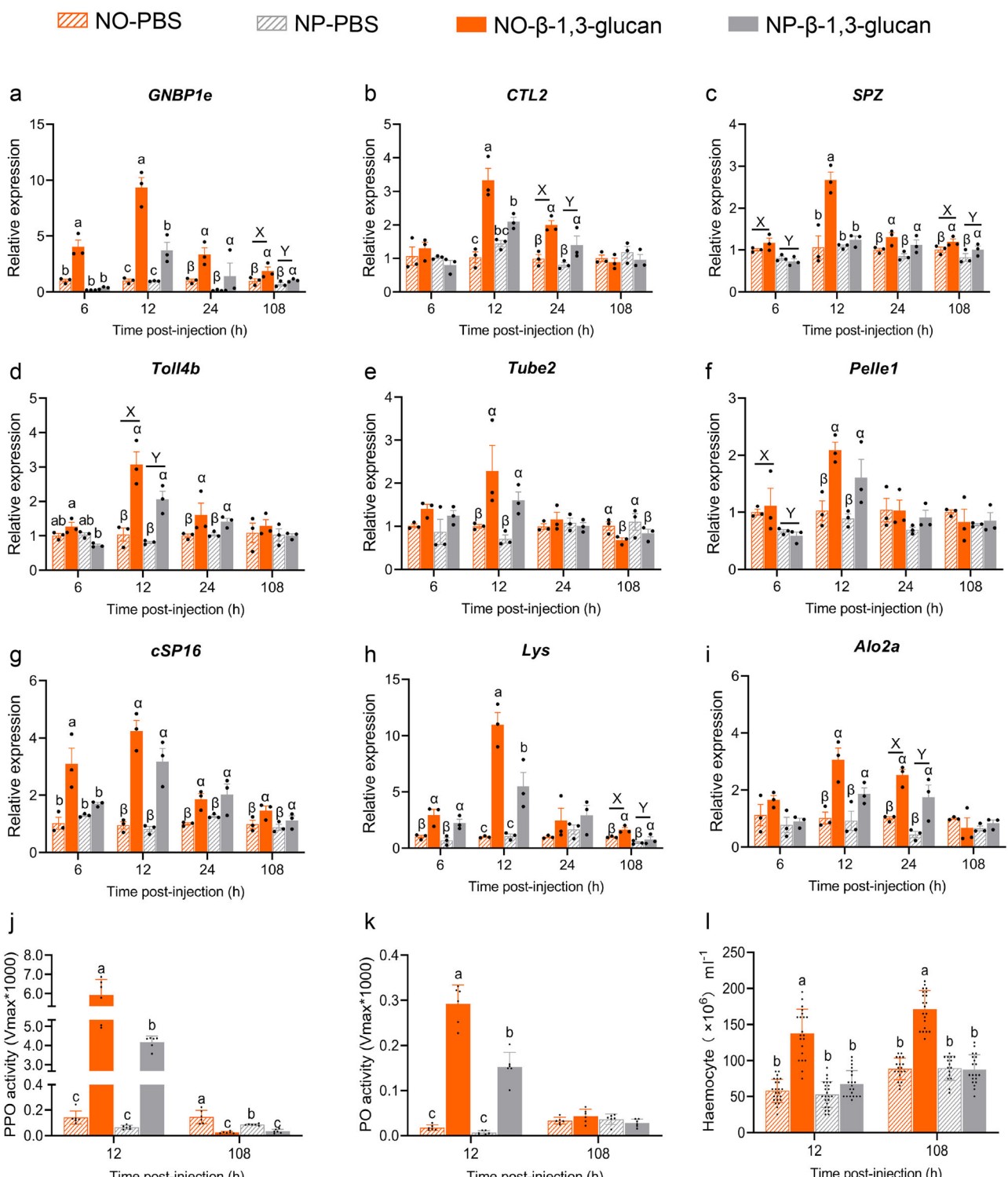

**Fig. 5 | β-1,3-glucan injection reveals nymphal diet-dependent differences in adult immune activation capacity.** 1.5-days-old adults developed from nymphs fed on an omnivorous diet (NO) or a phytophagous diet (NP) were injected with β-1,3-glucan (3125 ng per individual) or an equal volume of sterile PBS. This injection time point was selected to align both the developmental stage and immune activation window with those of the *Beauveria bassiana*-infected group, as *B. bassiana* typically penetrates the host cuticle within 24–48 h. Sampling was performed at 6 h, 12 h (developmentally corresponding to 2 dpi in the *B. bassiana* group), 24 h, and 108 h (corresponding to 6 dpi) post-injection to assess representative immune-related gene expression (*n* = 3 biologically independent samples) (**a–i**) and physiological immune parameters (**j–l**; PPO activity, PO activity, and hemocyte count) (PPO activity: *n* = 6 biologically independent samples; PO activity: *n* = 6 biologically independent samples; hemocyte count: *n* = 16–24 biologically independent samples). Data are presented as means ± SEM. Statistical annotations: different lower-case letters (a–d) indicate significant differences among all treatment combinations when a significant interaction effect was detected; uppercase letters (X, Y) indicate significant differences between diet treatments; Greek letters (α, β) indicate significant differences between infection conditions (Ctrl vs. Bb) ($P < 0.05$).

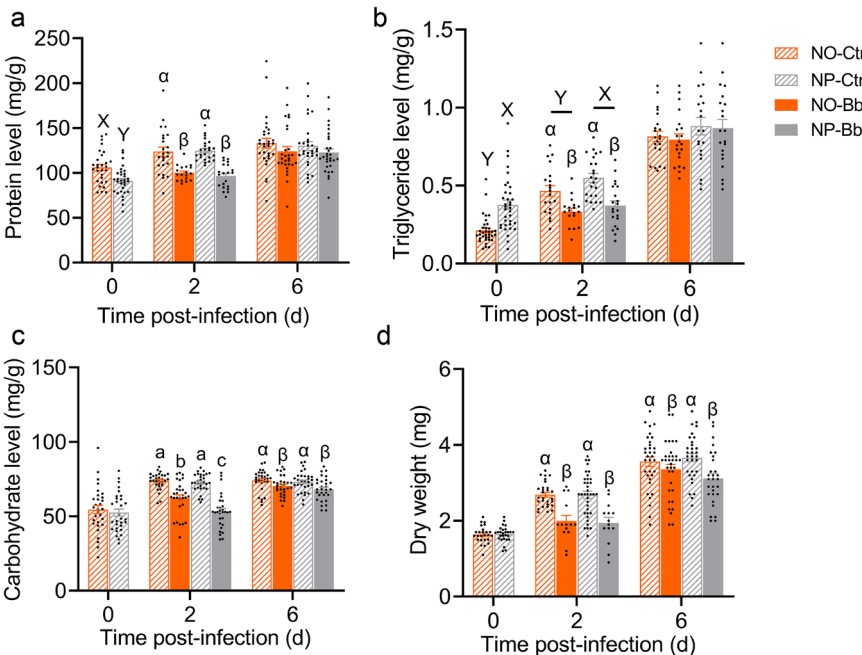

**Fig. 6 | Dynamic fluctuations of adult nutritional reserves under different nymphal diet treatments and pathogen exposure.** Nutritional reserve parameters include protein (**a**), triglyceride (**b**), carbohydrate (**c**), and dry weight (**d**) (protein: n = 18–31 biologically independent samples; triglyceride: n = 18–35 biologically independent samples; carbohydrate: n = 30 biologically independent samples; dry weight = 14–36 biologically independent samples). NO: adults developed from nymphs fed on an omnivorous diet; NP: adults developed from nymphs fed on a phytophagous diet; Bb: newly-emerged adults infected with *Beauveria bassiana*, Ctrl: newly-emerged adults treated with Tween 80. The data are presented as means ± SEM. Statistical annotations: different lowercase letters (a–d) indicate significant differences among all treatment combinations when a significant interaction effect was detected; uppercase letters (X, Y) indicate significant differences between diet treatments; Greek letters (α, β) indicate significant differences between infection conditions (Ctrl vs. Bb) ($P < 0.05$).

nutrient metabolism-related genes, except for *GlyS59* and *GCL3*, were significantly upregulated in response to infection. Conversely, in the NP group, only a subset of genes, including *HK2*, *PFK*, *KGDH3*, *Scs2*, *Gpat4a*, *HADH1*, *GOGAT1d*, *ALT2*, *GCL3*, and *PAH1*, were significantly upregulated upon infection. Interestingly, *PLOD* expression was significantly downregulated in the NP group post-infection (Figs. 7–9b and Supplementary Data 1). At 6 dpi, diet- and pathogen-induced gene expression changes were reduced, with only a limited number of nutrient metabolism-related genes remaining responsive to infection and nymphal diet. Specifically, *GAPDH1*, *IDH2*, and *FAS1b* were significantly downregulated by infection in both dietary groups, while *GlyS59*, *Acsl3*, and *HADH1* were significantly downregulated only in the NO group. In contrast, *GOGAT1d* and *PAH1* were significantly upregulated in response to infection regardless of dietary treatment, while *PK1* was significantly upregulated only in the NP group following infection. Additionally, the expression levels of *GCL3*, *PLOD*, *Gpat4a*, and *Lipase1f* appeared to be primarily influenced by dietary treatment rather than infection, with the NO group consistently displaying higher expression levels than the NP group (Figs. 7–9b and Supplementary Data 1).

## Discussion

Early-life nutritional environments are known to profoundly shape adult fitness[2], yet their enduring impact on adult immune function remains unclear. Immune traits often respond inconsistently to dietary variation; for example, in *Spodoptera littoralis*, protein-rich diets enhanced lytic and PO activity, whereas immune gene expression showed less predictable trends[26,27]. These discrepancies highlight the limitations of assessing immune capacity based on single traits. To reduce such bias, we evaluated immune capacity at multiple levels—survival after *B. bassiana* infection, hemocyte count, PO/PPO activity, and immune-related gene expression patterns profiled by transcriptomics and validated by RT-qPCR—providing an integrative view of how nymphal diet shapes adult immune capacity. We found that adults derived from nymphs reared on an omnivorous diet (NO) exhibited consistently higher survival and immune performance than those from a phytophagous diet (NP), demonstrating that early dietary conditions can durably shape adult immune function.

A central finding is that newly-emerged NO adults (0 dpi, prior to pathogen exposure) exhibited significantly higher baseline PPO activity and hemocyte counts than NP adults (Fig. 4). As hallmark components of constitutive immunity[13,28], their elevations suggest that early-life omnivory

confers pre-formed immune advantages, likely reflecting greater developmental investment in immunity driven by higher protein intake during the nymphal stage (Figs. 1a–d, 4, and 6). This interpretation is well supported by previous studies showing that protein-rich diets increases baseline PO activity and hemocyte counts in *Spodoptera littoralis*[29] and *Galleria mellonella*[30]. However, this advantage waned in the absence of infection: although PPO activity and hemocyte counts remained numerically higher in NO adults than in NP adults at 2 and 6 dpi under control conditions, these differences were no longer statistically significant (Fig. 4). Because all adults were maintained on the same phytophagous diet post-emergence, this attenuation likely reflects convergent nutritional inputs that diminished the developmental diet-driven immune advantages observed at emergence.

Despite sharing an identical phytophagous diet during adulthood, NO adults consistently mounted stronger inducible immune responses than NP adults. Upon *B. bassiana* infection, NO adults showed significantly elevated PPO/PO activity, hemocyte counts, and broad transcriptional upregulation of immune-related genes, most pronounced at 2 dpi—the major phase of immune activation in our sampling framework (Figs. 3 and 4 and Supplementary Data 2). This timing coincides with the 24–48 h window when *B. bassiana* conidia are reported to breach the cuticle in other insects[24,25], likely explaining the peak immune activation at 2 dpi.

The availability of nutritional resources is critical for immune activation, with dietary protein supporting hemocyte proliferation, melanization, and AMP production in many insects[10,11]. In our study, newly-emerged NO adults (0 dpi) displayed significantly higher protein content than NP adults (Fig. 6a), suggesting that nutrient accumulation during the nymphal stage may directly support immune activation in adulthood. By 2 dpi, however, protein content had equalized between groups under the common phytophagous diet (Fig. 6), suggesting that the enhanced immunity of NO adults is unlikely to result solely from adult nutrient stores and may involve additional mechanisms beyond nutrient accumulation.

Another mechanism underlying the stronger inducible immunity of NO adults may be the developmental establishment of a larger baseline hemocyte pool. In *Locusta migratoria*, circulating hemocytes proliferate and serve as the primary source of new hemocytes after infection[31]. Based on this, we speculate that a larger baseline circulating hemocyte reservoir in NO adults may allow faster and more robust clonal expansion upon challenge. Consistent with this, *Drosophila melanogaster* populations evolving under

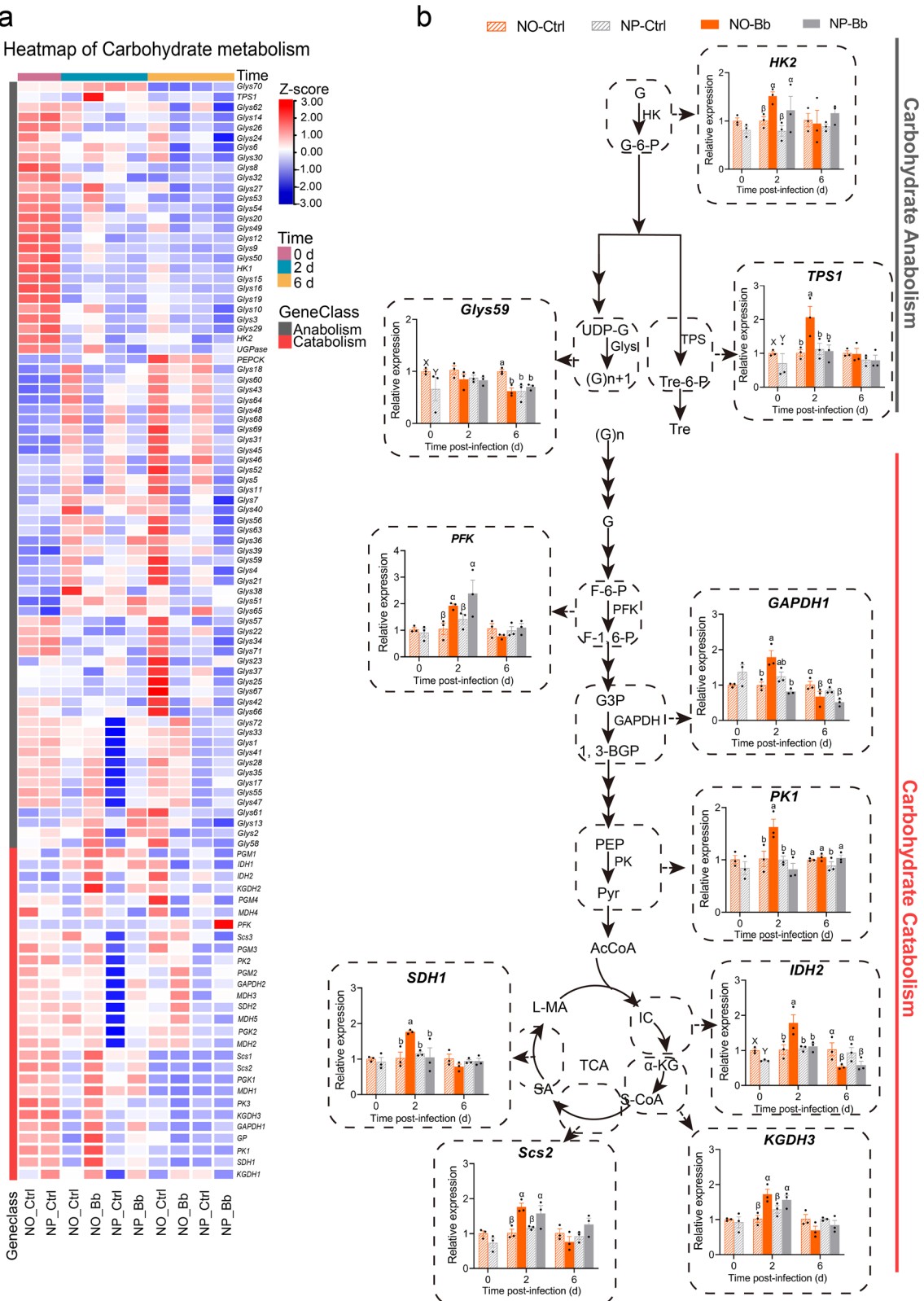

**Fig. 7 | Effect of nymphal diet on the expression patterns of carbohydrate metabolism-related genes in adults. a** The heatmap of clustered tBLASTn for differentially expressed genes linked to carbohydrate anabolism and catabolism. Color scale represents *Z*-scores computed by standardizing the expression of each gene across all samples. Red indicates expression higher than the average level of that gene across samples (*Z* > 0), blue indicates lower expression (*Z* < 0), and white indicates near-average levels (*Z* ≈ 0). **b** Validation of the expression of key genes involved in carbohydrate anabolism and catabolism by RT-qPCR. *n* = 3 biologically

independent samples. NO: adults developed from nymphs fed on an omnivorous diet; NP: adults developed from nymphs fed on a phytophagous diet; Bb: newly-emerged adults infected with *Beauveria bassiana*, Ctrl: newly-emerged adults treated with Tween 80. The data are presented as means ± SEM. Statistical annotations: different lowercase letters (a–d) indicate significant differences among all treatment combinations when a significant interaction effect was detected; uppercase letters (X, Y) indicate significant differences between diet treatments; Greek letters (α, β) indicate significant differences between infection conditions (Ctrl vs. Bb) (*P* < 0.05).

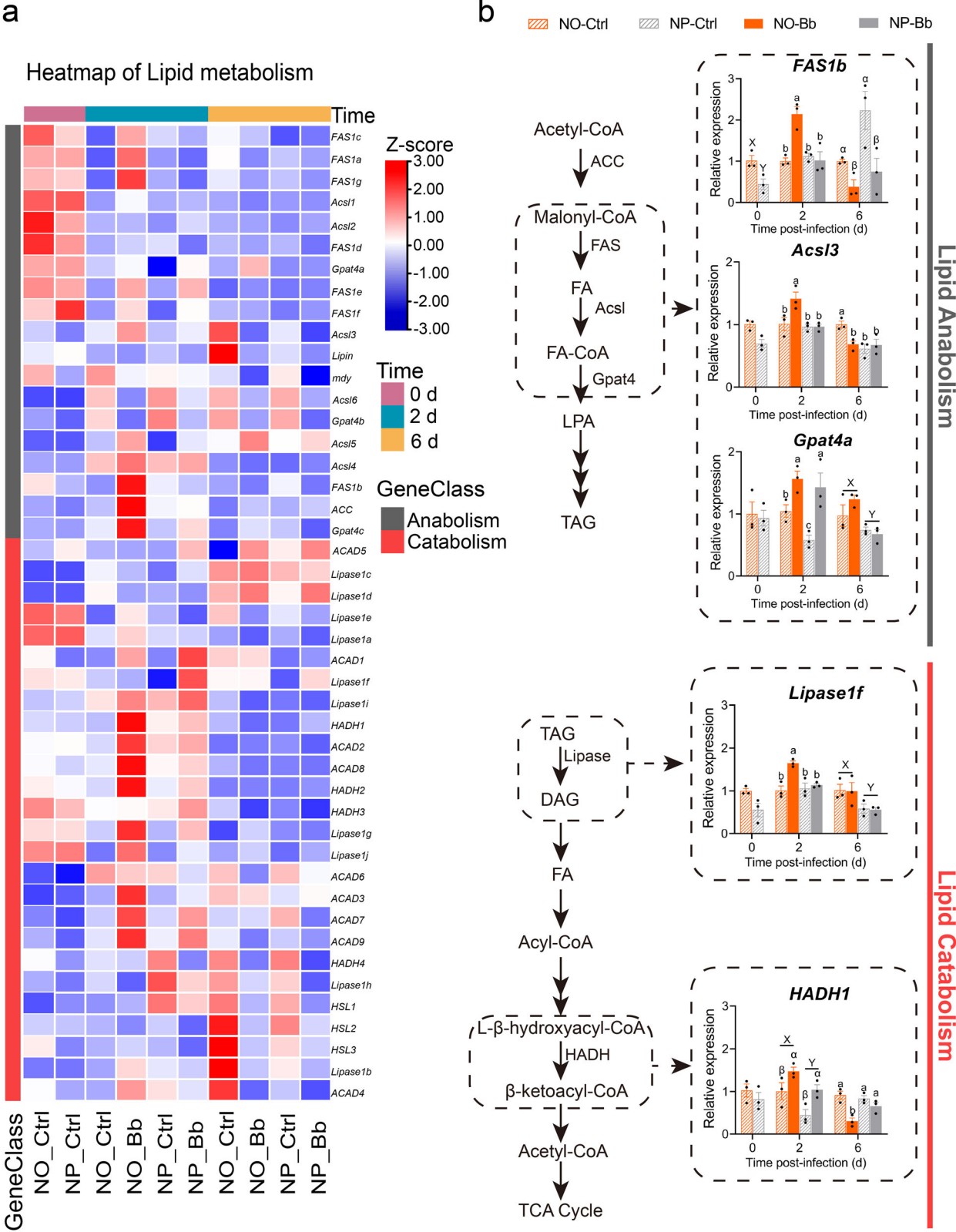

**Fig. 8 | Effect of nymphal diet on the expression patterns of lipid metabolism-related genes in adults. a** The heatmap of clustered tBLASTn for differentially expressed genes linked to lipid anabolism and catabolism. Color scale represents Z-scores computed by standardizing the expression of each gene across all samples. Red indicates expression higher than the average level of that gene across samples ($Z > 0$), blue indicates lower expression ($Z < 0$), and white indicates near-average levels ($Z ≈ 0$). **b** Validation of the expression of key genes involved in lipid anabolism and catabolism by RT-qPCR. $n = 3$ biologically independent samples. NO: adults developed from nymphs fed on an omnivorous diet; NP: adults developed from nymphs fed on a phytophagous diet; Bb: newly-emerged adults infected with *Beauveria bassiana*, Ctrl: newly-emerged adults treated with Tween 80. The data are presented as means ± SEM. Statistical annotations: different lowercase letters (a–d) indicate significant differences among all treatment combinations when a significant interaction effect was detected; uppercase letters (X, Y) indicate significant differences between diet treatments; Greek letters (α, β) indicate significant differences between infection conditions (Ctrl vs. Bb) ($P < 0.05$).

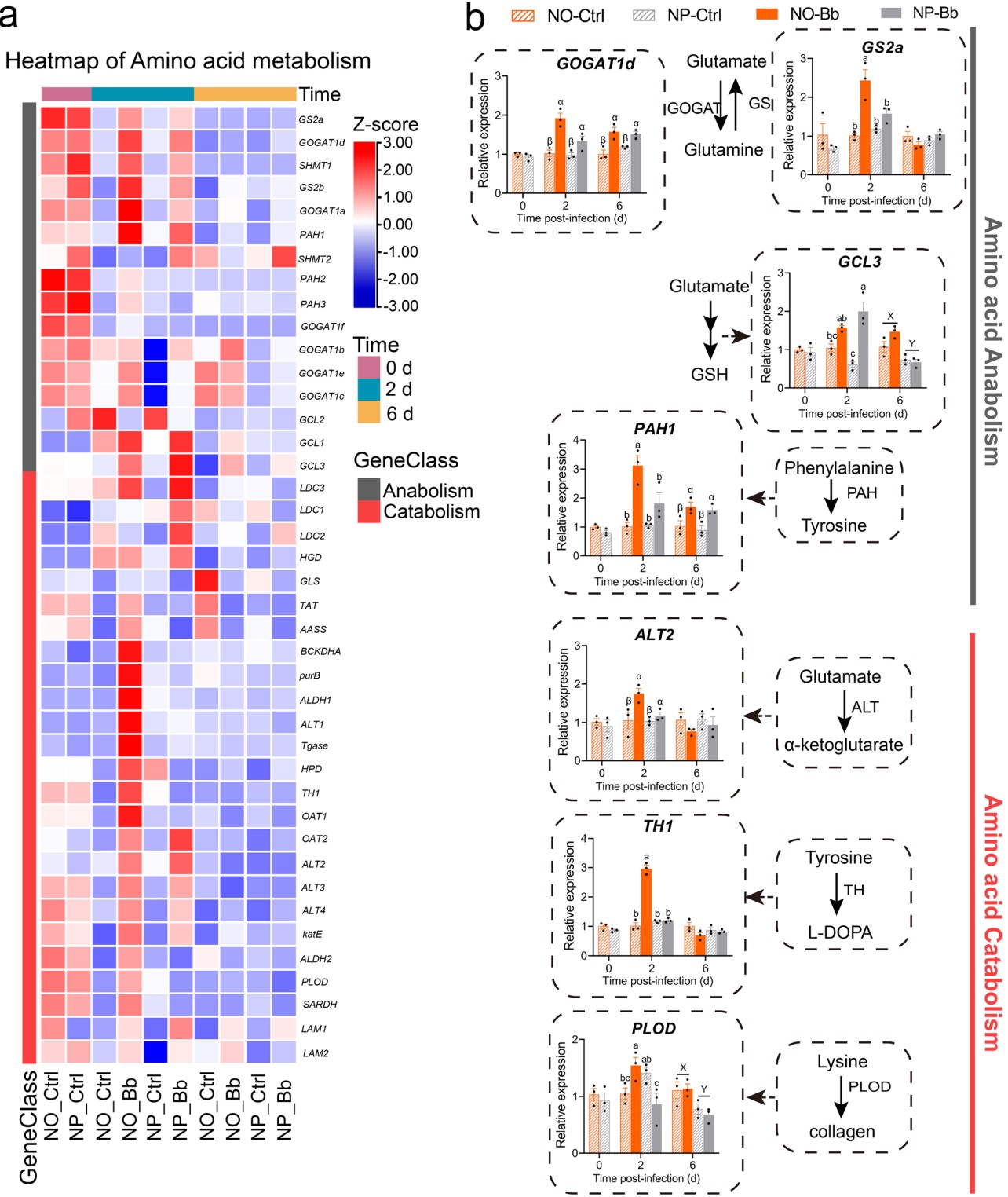

**Fig. 9 | Effect of nymphal diet on the expression patterns of amino acid metabolism-related genes in adults. a** The heatmap of clustered tBLASTn for differentially expressed genes linked to amino acid anabolism and catabolism. **b** Validation of the expression of key genes involved in amino acid anabolism and catabolism by RT-qPCR. $n = 3$ biologically independent samples. NO: adults developed from nymphs fed on an omnivorous diet; NP: adults developed from nymphs fed on a phytophagous diet; Bb: newly-emerged adults infected with

*Beauveria bassiana*, Ctrl: newly-emerged adults treated with Tween 80. The data are presented as means ± SEM. Statistical annotations: different lowercase letters (a–d) indicate significant differences among all treatment combinations when a significant interaction effect was detected; uppercase letters (X, Y) indicate significant differences between diet treatments; Greek letters (α, β) indicate significant differences between infection conditions (Ctrl vs. Bb) ($P < 0.05$).

high parasitism pressure exhibited higher baseline hemocyte counts and also produced more hemocytes upon infection[14].

Hemocytes are key mediators of insect immune responses, executing cellular defenses such as phagocytosis, nodulation, and encapsulation, and serving as the principal site for PPO synthesis while also contributing to the production of immune effectors, including AMPs and recognition receptors[32–34]. In *Drosophila*, AMP expression in hemocytes can even exceed that in fat body[35]. In our study, NO adults challenged with *B. bassiana* showed parallel increases in hemocyte counts, PPO/PO activity, and immune-related gene expression—most pronounced at 2 dpi (Figs. 3 and 4 and Supplementary Data 2)—indicating a coordinated enhancement of inducible immunity. Similar associations between hemocyte expansion and enhanced immune effector production and activity have been reported in *Glyphodes pyloalis*[36] and *Apis mellifera*[37]. Given the roles of hemocytes in PPO and immune effector production, we propose that increased hemocyte counts may enhance both PPO/PO-mediated melanization and the transcriptional activation of antifungal defenses. The fat body, another major site of AMP production[38,39], may also contribute to dietary effects on immunity, but its role remains unresolved.

Coordinated elevations of PPO/PO activity, hemocyte counts, and immune-related gene expression in NO adults provide a direct functional basis for their superior resistance to *B. bassiana* (Figs. 3 and 4). Transcriptomic analysis revealed that, compared with NP-Bb adults, the immune-related genes upregulated in NO-Bb adults were enriched for fungal recognition, the Toll pathway, and downstream antifungal effectors such as Alo and lysozymes (Fig. 3), indicating a targeted reinforcement of antifungal defenses. Elevated PPO/PO supply likely accelerates melanization, generating quinones and melanin that encapsulate fungal hyphae and restrict colonization[40]. In parallel, higher hemocyte counts facilitate rapid aggregation at infection sites and, together with melanization, promote nodule formation and pathogen clearance[41]. Antifungal peptides further contribute by suppressing fungal growth[42]. Collectively, these processes likely underpin the higher survival of NO adults under fungal challenge.

Mounting an effective immune response requires not only immune tissues and effectors but also sufficient metabolic support. Growing evidence shows that carbohydrate, lipid, and amino acid metabolism are regulated by, and also contribute to, insect immunity[43–48]. In line with this coupling, NO-Bb adults exhibited higher expression of genes for lipid catabolism (*Lipase1f* for lipolysis, *HADH1* for β-oxidation)[49], glycolysis (*GAPDH1*, *PK1*), and the TCA cycle (*SDH1*, *IDH2*)[50] than NP-Bb or uninfected controls. These pathways are known to provide energy for hemocyte proliferation, phagocytic activity, and effector synthesis[43,44,50,51]. Extending beyond carbon and lipid metabolism, NO-Bb adults also upregulated multiple amino acid metabolism genes relevant to immune function. Among these, *PAH1* and *TH1* convert phenylalanine into tyrosine and subsequently DOPA—the critical substrates for PPO/PO-mediated melanization[40,52], while *GOGAT1d*, *GS2a*, *GCL3*, and *ALT* participate in glutamate, glutamine, and glutathione metabolism (Fig. 9). These metabolites enhance immune competence through distinct mechanisms: glutamate promotes lipid mobilization to fuel immune activation[53], glutamine uptake by phagocytes feeds into the TCA cycle via conversion to α-ketoglutarate, thereby sustaining phagocytic activity[54], and glutathione (GSH) synthesized via *GCL3* mitigating oxidative stress during infection[55,56]. Together, these results suggest that early-life omnivory primes a more efficient metabolic infrastructure supporting the energetic and biosynthetic demands of inducible immunity. Interestingly, despite mounting stronger immune responses, NO adults showed milder depletion of protein, triglycerides, carbohydrates, and dry mass by 2 dpi compared with NP adults (Fig. 6), suggesting that this enhanced metabolic capacity may allow immune activation to be sustained with reduced nutrient loss.

Together, our findings outline a mechanistic framework in which early-life omnivory durably enhances adult immune capacity through complementary pathways. Juveniles fed an omnivorous diet accumulated more protein, a resource that likely supported immune tissue development, PPO production, and AMP synthesis[27], although this nutritional advantage diminished once adults shifted to a uniform phytophagous diet. Early omnivory also promoted greater structural investment into immune systems, reflected by elevated hemocyte counts at emergence, which likely supported hemocyte proliferation and immune effector production during infection. Finally, early nutritional experience appeared to establish a more robust metabolic infrastructure, enabling adults to meet the energetic and biosynthetic demands of inducible immunity. Collectively, these effects support the hypothesis of developmental programming, wherein early-life dietary conditions durably shape adult immune competence. Such programming may arise from structural reinforcement of immune tissues, accumulation of enzymatic precursors, and enhanced metabolic or signaling responsiveness under pathogen challenge.

Our findings can be contextualized within two existing hypotheses on how developmental nutrition shapes adult immunity. One posits that improved food quality enhances overall body condition in a non-specific manner, thereby boosting all adaptive traits, including immunity, as shown in *Lestes viridis* and *Aedes aegypti*, where restricted juvenile nutrition impaired adult immunity alongside reduced adult body size and nutritional reserves[57–59]. Another posits that early-life diet alters the proportion of resource allocation to immunity, independent of overall body condition, through investment in immune tissues or increased supply of immune-limiting nutrients. For example, *D. melanogaster* adults from larvae reared on a low-protein diet exhibit reduced immune-related genes expression despite similar adult body weight to those from a high-protein diet[60]. In our system, both routes appear to operate: NO adults showed targeted immune investment at emergence, with higher baseline hemocyte counts and PPO activity (Fig. 4), while also displaying broader transcriptional activation of carbohydrate, lipid, and amino acid metabolism (Figs. 7–9), suggesting an overall enhancement in physiological capacity. These results also highlight the limitations of equating overall body condition with simple proxies such as body weight or dry mass, since immune performance depends not only on nutrient reserves but also on their mobilization efficiency[43,51,53,61].

Notably, at 0 dpi—before pathogen exposure—we observed a marked upregulation of immune-related genes in both NO and NP adults, which declined sharply by 2 and 6 dpi in uninfected controls (Fig. 3a). Additional analysis confirmed that this early activation was not attributable to Tween 80 treatment, as untreated and Tween-treated adults showed no significant differences at 0 or 2 dpi (Supplementary Fig. 3). Instead, the transient response likely reflects prophylactic immunity associated with adult emergence, when the cuticle is newly formed and vulnerable to microbial invasion[62]. Similar short-lived immune activation has been reported in *Ae. aegypti*[62] and *D. melanogaster*[63]. Interestingly, this transcriptional pulse was not mirrored by changes in PPO activity or hemocyte counts (Fig. 4), consistent with distinct activation kinetics among immune components: PPO/PO-mediated melanization and hemocyte-driven phagocytosis are typically rapid inducible upon infection, while AMPs synthesis is often slower[28,64] and may require earlier transcriptional priming.

Beyond macronutrients, natural diets also contain micronutrients, secondary metabolites, and microorganisms, which can modulate insect-pathogen interactions through various mechanisms[65–68]. Whether such factors acquired during development exert lasting effects on adult immunity remains unclear. Moreover, while our findings establish a strong association between early-life diet and adult immunity, causality remains to be established. Future studies employing functional approaches—such as RNA interference of immune or metabolic regulators, nutrient-specific manipulation during development, or targeted perturbation of microbial communities—will be essential to disentangle the mechanisms linking juvenile nutrition to adult immune function.

In summary, our data show that early-life omnivory is strongly associated with enhanced adult immune competence in a mirid bug, supported by convergent evidence across survival, cellular, enzymatic, and transcriptomic readouts. Taken together, our results offer a developmentally grounded framework linking early nutritional environments with adult immune function, contributing to an integrated understanding of nutritional and developmental immunology.

## Methods

### Source of insect and entomopathogenic fungus

*A. suturalis* were initially obtained from Hubei Insect Resources Utilization and Sustainable Pest Management Key Laboratory, Huazhong Agricultural University, Wuhan, China. The colonies were reared as described in Luo et al.[20]: *A. suturalis* were reared at $25 \pm 1\ °C$ under $75 \pm 5\%$ relative humidity with a 16:8 h light: dark photoperiod. Nymphs and adults were reared in plastic cages (225 mm long, 150 mm wide, 110 mm high) at a density of approximately 60 individuals per cage. Nymphs were fed fresh mung bean sprouts, and adults were fed fresh green beans. All food was supplied daily in sufficient quantities. In addition, aphids (*Acyrthosiphon pisum*) were provided daily in sufficient amounts as a dietary supplement for both nymphs and adults. Newly-emerged adults within 0–12 h were defined as 0-day-old, each additional 24 h adds one day of age.

*B. bassiana* isolate 3083 (obtained from the Anhui Provincial Key Laboratory of Microbial Control, School of Forestry & Landscape Architecture, Anhui Agricultural University, Hefei, China) was grown and maintained on Potato Dextrose Agar (PDA) medium in petri dishes (90 mm diameter) at 25 °C for 15 days. A 0.05% (v/v) sterile Tween 80 was used to collect the fungal conidia and adjusted to a concentration of $2 \times 10^{-7}$ conidia $mL^{-1}$ for subsequent experiments.

### Dietary and infection treatments

Newly hatched *A. suturalis* nymphs were individually placed in 60 mm diameter petri dishes and randomly assigned to one of two dietary treatment groups: phytophagous and omnivorous. In the phytophagous treatment group, nymphs were solely fed mung bean sprouts (NP). Conversely, in the omnivorous treatment group, nymphs were allowed to feed freely on both mung bean sprouts and aphids (NO). Upon adult emergence, 0-day-old adults from each dietary treatment were selected and immersed in a conidial suspension of *B. bassiana* (Bb) or a sterile Tween 80 solution (Ctrl) for 20 s. Subsequently, all individuals were reared exclusively on mung bean sprouts (Fig. 1a). The rest of the feeding conditions were the same as above. Survival was monitored daily for eight days. More than 30 individuals from each treatment group were evaluated per biological replicate, with a total of three biological replicates conducted.

### Sample collection

To monitor changes in immune responses and physiological status during pathogen challenge, we collected samples from both NP- and NO-treated adults at three defined time points based on infection progression dynamics: 0 days post-infection (dpi), representing the pre-infection baseline, when adults from both dietary treatments had undergone the same Tween 80 treatment; 2 dpi, corresponding to the early infection phase (~ 10% mortality in the NP-Bb group); and 6 dpi, representing the mid-to-late stage of infection progression (~50% mortality in the NP-Bb group). In total, ten sample sets were generated for subsequent analyses, namely 0d_NP_Ctrl, 0d_NO_Ctrl, 2d_NP_Ctrl, 2d_NO_Ctrl, 2d_NP_Bb, 2d_NO_Bb, 6d_NP_Ctrl, 6d_NO_Ctrl, 6d_NP_Bb, and 6d_NO_Bb.

### Transcriptome sequencing

For each treatment and time point, three biological replicates were prepared, each biological replicate consisting of three individuals pooled, homogenized, and processed as a single sample for RNA extraction and subsequent analysis. The total RNA was extracted using RNAiso Plus reagent (TaKaRa, Kyoto, Japan) following the manufacturer's protocol. RNA integrity was evaluated on 1.5% agarose gels and quantified using a Nano-Drop 2000 (Thermo Scientific, Wilmington, DE, USA). Using an Illumina TruseqTM RNA sample prep Kit (Illumina, San Diego, CA, USA), two µg RNA samples were used to construct cDNA libraries. RNA sequencing was performed on an Illumina Novaseq 6000 system (Illumina, San Diego, CA, USA). Raw reads were subjected to quality assessment using Fastp v0.19.5. Each sample retained approximately 41.5–59.8 million clean reads, with Q30 > 97.6% and clean rate > 93.5%. The resulting clean reads were then de novo assembled using Trinity v2.8.5, generating 152,451 transcripts with an N50 of 1873 bp and a mean transcript length of 1077 bp. Structurally unsupported or low-confidence transcripts were filtered using TransRate v1.0.3, and redundant sequences were clustered using CD-HIT v4.5.7 to generate unigenes. Assembly completeness was assessed using BUSCO v3.0.2 with the insecta_odb10 database, which showed 96.7% complete (57.6% Single-copy; 39.1% duplicate). Unigenes were annotated by querying six public databases (NR, Swiss-Prot, Pfam, COG, GO, and KEGG) to support gene function and pathway analysis. Homology searches against the NR, Swiss-Prot, and COG databases were performed using DIAMOND v0.9.24 ($E$-value $< 1e^{-5}$). Protein domains were identified using HMMER 3.2.1 against the Pfam-A database. GO terms were assigned using Blast2GO v6.0.3, and KEGG pathway mapping was performed via the KAAS online tool (https://www.genome.jp/tools/kaas/). Gene expression levels were quantified with RSEM v1.3.3, and differential expression analysis was performed using DESeq2 v1.24.0 ($p$-adjust $< 0.001$, fold change $> 2$).

### Identification of immune-related genes

A reference set of insect immune-related genes was compiled from NCBI, incorporating well-annotated sequences from both model organisms and hemipteran pests, including *Drosophila melanogaster*[69], *Anopheles gambiae*[70], *Apis mellifera*[71], *Bombyx mori*[72], *Tribolium castaneum*[73], *Nilaparvata lugens*[74], *Aphis gossypii*[75], and *Halyomorpha halys*[76]. These amino acid sequences were used as queries in a tBLASTn search against the *A. suturalis* unigene dataset using Basic Local Alignment Search Tool v2.16.0, with an $E$-value cutoff of $1e^{-5}$. The resulting hits were manually curated and confirmed by domain architecture using CD-search against the NCBI Conserved Domain Database (CDD). Only unigenes containing conserved domains consistent with known immune-related gene families were retained.

### Selection of candidate genes for RT-qPCR validation

Candidate genes for RT-qPCR validation were chosen from the differentially expressed gene (DEG) sets based on three criteria: pathway relevance, statistical significance, and functional annotation. For immune-related genes, we focused on key components of inducible antifungal immune pathways that were significantly enriched in the transcriptomic analysis, including fungal recognition factors, core components of the Toll signaling pathway, and antifungal effectors. For nutrient metabolism-related genes, we selected central nodes in anabolic or catabolic pathways that are functionally linked with immunity. When multiple members of a gene family met these criteria, we prioritized those showing the largest differential expression between NO-Bb and NP-Bb groups, higher sequence conservation, complete functional domains, and well-supported annotation.

### Validation of gene expression by RT-qPCR

One µg RNA samples were reverse transcribed using the PrimeScript™ RT Master Mix (Yeasen, Shanghai, China). RT-qPCR was performed with a CFX Connect Real-Time System (Bio-Rad, Hercules, CA, USA) using MonAmp™ SYBR® Green qPCR Mix (Monad Biotech, Wuhan, China) in a volume of 10 µL (2 µL of 20-fold dilution cDNA template, 1 µL of 400 nmol $L^{-1}$ of each gene-specific primer, 5 µL of $2 \times$ MonAmp™ SYBR® Green qPCR Mix). The PCR was performed under the following conditions: 95 °C for 3 min, followed by 40 cycles of 95 °C for 10 s and 60 °C for 30 s. For each treatment and time point, three biological replicates were analyzed, with each replicate consisting of three pooled individuals. Technical triplicates were performed for each biological sample. The RT-qPCR data were collected and analyzed via the $2^{-\Delta\Delta Ct}$ method[77]. All primers were validated to have amplification efficiencies between 90% and 110%, meeting the accepted criteria for applying this method. The primers used in this study are shown in Supplementary Table 1. The *RPS15* gene was used as a reference for gene expression normalization[22].

### Hemocyte quantification

Hemolymph was collected from the adult by excising the legs and drawing out the exuding hemolymph using a thin glass capillary. The collected

hemolymph was subsequently used for hemocyte counting, following the method outlined by Lackie[78]. Briefly, 0.5 μL of the hemolymph sample was gently mixed with 49.5 μL filtered, cold phosphate-buffered saline (PBS, 10 mM, pH 7.4). A 10 μL aliquot of this mixture was then pipetted onto a hemocytometer (Shanghai Qiujing Biochemical Reagent Instrument Co. Ltd., China) and examined under a microscope (Leica Microsystems, Wetzlar, Hesse, Germany) at 400× magnification to evaluate the hemocyte count[79,80]. At least 15 biological replicates were carried out.

### The activity of phenoloxidase and prophenoloxidase

The enzymatic activity of phenoloxidase (PO) and prophenoloxidase (PPO) was assessed spectrophotometrically following the method described in a previous study[81]. Briefly, 1 μL hemolymph sample was gently mixed with 20 μL cold PBS, centrifuge at $4000 \times g$ for 15 min at 4 °C and collect the supernatant. Then, 5 μL of the supernatant was loaded into a microplate well containing 160 μL of PBS, and 20 μL of L-DOPA solution (4 mg/mL) (Coolaber, Beijing, China) to quantify PO activity. We replaced the PBS with chymotrypsin (2 mg/mL) (Coolaber, Beijing, China) to measure the total PO activity. The enzymatic reaction was allowed to proceed at 30 °C for 40 min in a microplate reader (SpectraMax 190, Molecular Devices, USA), with readings taken every 30 s at 490 nm. Enzyme activity was calculated as the slope (V max) of the reaction curve during its linear phase. PPO activity was determined by subtracting PO activity from total PO activity[82]. All measurements were conducted in triplicate, with six biological replicates performed for each treatment.

### Immune elicitor injection assay

To directly assess whether nymphal diet shapes the adult capacity for immune activation, we performed an immune elicitation assay using β-1,3-glucan, a fungal-specific immune inducer[23]. 1.5-days-old adults from the NO and NP groups were injected with 3125 ng of β-1,3-glucan (Sigma-Aldrich, St. Louis, MO, USA) dissolved in 100 nL of sterile phosphate-buffered saline (PBS) into the hemocoel using a micro-injector (World Precision Instruments, Sarasota, FL, USA). This injection time point was selected to align their developmental stage and timing of immune activation with those of the *B. bassiana*-infected group, considering that *B. bassiana* typically penetrates the host cuticle within 24–48 h[24,25]. Individuals injected with an equal volume of sterile PBS served as controls. The dietary treatments and rearing conditions were identical to those used in the *B. bassiana* infection experiment. Samples were collected at 6 h, 12 h (developmentally corresponding to 2 dpi in the *B. bassiana* group), 24 h, and 108 h (corresponding to 6 dpi in the *B. bassiana* group) post-injection for analyses of PPO and PO activities, total hemocyte counts, and qRT-PCR quantification of selected immune-related genes. These genes were the same as those validated in the *B. bassiana* infection experiment.

### Measurements of nutrient reserve

**Measurements of protein and triglycerides content.** Total triglyceride and protein content were quantified using Triglycerides Assay Kit (Nanjing Jiancheng Institute, Nanjing, China) and BCA Protein Assay kit (Biosharp, Hefei, China), respectively. According to the manufacturer's instructions, each individual was weighed using a microbalance (MS205DU, Mettler Toledo, Switzerland) to obtain fresh body mass with an accuracy of ± 0.0001 mg. A PBS solution was then added to each sample at a weight-to-volume ratio of 1 g:10 mL. Then, these samples were homogenized and centrifuged at 2500 rpm for 10 min at 4 °C. The resulting supernatant was divided into two portions: one for triglyceride content and the other for total protein content analysis. For triglyceride quantification, 2.5 μL of the supernatant was combined with the reaction solution and incubated at 37 °C for 10 min, after which absorbance was measured at 500 nm. For protein content determination, 1 μL of the supernatant was mixed with 19 μL of PBS, and 20 μL of this mixture was added to 200 μL of BCA reaction solution. The mixture was incubated at 37 °C for 10 min, with absorbance detected at 562 nm. Absorbance

readings were performed at room temperature using a microplate reader (SpectraMax 190, Molecular Devices, USA) with the Soft Max Pro software. More than 30 individuals were tested in each treatment.

**Measurements of total carbohydrate content.** Total carbohydrate content was measured using the Total Carbohydrate Content Assay Kit (Solarbio, Beijing, China). A pool of five individuals was homogenized in 1250 μL of homogenization medium, comprising 500 μL of Reaction Solution 1 and 750 μL of distilled water, and then boiled for 30 min. After cooling to room temperature, 500 μL of Reaction Solution 2 was added, followed by thorough mixing and centrifugation at $8000 \times g$ for 10 min. Subsequently, 150 μL of the supernatant was combined with 150 μL of Reaction Solution 3 and boiled at 100 °C for 10 min. Upon cooling to room temperature, the reaction mixture was diluted with 900 μL of distilled water, and absorbance was measured at 540 nm using spectrophotometer UV-6300 (Mapada Instruments, Shanghai, China). Each treatment included at least six independent biological replicates and three technical replicates. Measurements were taken in triplicate, and any individuals for which the coefficient of variation across these replicates exceeded 30% were excluded from the analysis[81].

**Dry weight.** Estimates of the dry weight of individual adult were obtained from at least 30 samples, which were dried at 60 °C until a constant weight was achieved[83]. Each sample was weighed three times using a microbalance (MS205DU, Mettler Toledo, Switzerland), and the average of these measurements was calculated to determine the final dry weight.

### Data analysis

Survival curves were analyzed using the Kaplan–Meier method and compared statistically using the log-rank test. For data involving two groups, the Two-tailed Student's unpaired *t*-test was employed. Comparisons among multiple groups were conducted using two-way ANOVA followed by Tukey's post-hoc test. The Shapiro–Wilk test was utilized to assess the normality of distributions, while the Levene test was applied to evaluate equality of variance. Data analysis was performed using SPSS 19.0, and figures were created with GraphPad Prism 8 and compiled in Photoshop CS6.

### Statistics and reproducibility

Data are presented as means ± SEM. Statistical differences among groups were assessed using ANOVA with Tukey's multiple comparison test or *T*-test as appropriate. Survival curves were analyzed using the Kaplan–Meier method and compared statistically using the log-rank test.

Survival assays included 30–50 individuals per treatment and were performed with three biological replicates, with no technical replication. Biochemical assays (protein, lipid, glycogen, and carbohydrate quantification) and dry weight measurements were conducted using more than 15 individuals per treatment, with each individual treated as one biological replicate. Each biological replicate was measured in three technical replicates. For immune-related assays, PPO and PO enzymatic activities and hemocyte counts were measured from pooled hemolymph samples, with hemolymph collected from approximately 15 individuals per biological replicate. PPO and PO activities were assessed using six biological replicates per time point, whereas hemocyte counting was performed using at least 15 biological replicates. Measurements were taken in triplicate. Gene expression analyses (RT-qPCR) were performed using three biological replicates per treatment, each consisting of RNA extracted from three female adults. Technical triplicates were performed for each biological sample. All sample sizes and replicate numbers are reported in the relevant figure legends and "Methods" sections.

### Reporting summary

Further information on research design is available in the Nature Portfolio Reporting Summary linked to this article.

## Data availability

The source data for all statistical graphs are provided in Supplementary Data 3. The RNA-Seq data files generated in this study have been deposited in the NCBI Sequence Read Archive (SRA) under BioProject PRJNA1191641. The de novo transcriptome assembly has been deposited in the NCBI Transcriptome Shotgun Assembly (TSA) database under the accession number GLJW00000000.

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

## Acknowledgements

This project was funded by the National Natural Science Foundation of China (31901890, 32471571).

## Author contributions

J.L. and L.T.X. conceived the project. W.N.L., J.F.G., J.L., and L.T.X. wrote the manuscript. W.N.L., J.F.G., C.L., and Y.X.L. performed experiments, analyzed data, and/or prepared the manuscript. C.L., and Y.X.L. advised on the study, and/or helped with data analysis.

## Competing interests

The authors declare no competing interests.
