## [Transparent Peer Review file · Communications Biology]

Nymphal diets boost adults' immunity via strengthened constitutive immunity and metabolic capacity in *Adelphocoris suturalis*

Corresponding Author: Professor Jing Luo

Version 0:

Reviewer comments:

Reviewer #1

(Remarks to the Author)

The manuscript entitled "Nymphal diets boost adults' immunity via immune storage and enhanced metabolism in a Mirids bug" present interesting, new data concerning how the nymphal diet may affect the functioning of immune system of adult individuals. Generally, the manuscript is well written but some of the issues concerning research methodology and results should be clarified by Authors. For this reason, I recommend major revision. Detailed comments you can find below.

INTRODUCTION

Lines 57-74. I understand the Authors intention concerning division of immune system mechanisms into constitutive and induced immunity. However, if the article is directed to broader audience, Authors should also explain reader that insect immunity based mostly on the innate mechanisms etc.

Line 72. Some articles suggest that there is more than one lysozyme. Also, in several articles, lysozymes are wrongly classified as AMPs.

RESULTS

Lines 107-120. Why the survival experiments lasted only 8 days?

Line 115 and Line 116. Please add the SEM or SD value for the LT50.

Lines 121-125. The results related to the differences between diets may be separate from the results of the survival experiment. I think it is good to start with this information. Additionally, please explain why the lipid content and the fatty acid composition were not analysed? On the basis of the presented results, the Authors are aware of the role of lipids in energy metabolism and their direct and indirect roles in shaping insect immune system activity.

Fig 1C and D. The abbreviation of group should be consistent in whole manuscript. I suggest change the group names on the NO and NP.

Figures 1, 3, 4, 5, 6, 7 and 8. I think the plot style should be unified, including indication of statistically significant changes. In the current version, data are sometimes presented as bar plots, sometimes as scatter plots with bars, and sometimes as scatter plots. Additionally, sometimes in one graph, we see different styles of indication of statistically significant changes. Authors should choose whether the significant changes will be indicated as asterisks or different letters.

Lines 131-133 and Lines 471-476. Please explain why this three time-points were chosen?

Lines 169-199 and Figure 3. A comparison of the transcriptomic analysis results, which are presented as heatmaps, and the qPCR results raises some doubts. On the basis of the heatmap, we may assume that the use of Tween 80 as a control may substantially affect the expression levels of immune-related genes even two days after its application (compare NO vs NO-

Ctrl and NP vs NP-Ctrl). In addition, most of the qPCR analyses did not confirm the results of the transcriptomic data. I know that this is normal that may exist the differences between transcriptomic data and qPCR results which is related to the methodology of these techniques. However, in the case of some genes, the trends between the control individuals (NO-Ctrl and NP-Ctrl) and insects treated with *B. bassiana* were opposite, for example, in the case of *Lys* on day 6 (NP groups), in the case of *Toll4b* on day 2, or in the case of *Alo2a* on day 2 (NP groups). There are many such examples, so please carefully revised your data.

Lines 175-178. This part better fits the Discussion than the Results.

Line 204. Should be "hemocyte number" or "hemocyte count".

Lines 215-216. This sentence should be moved to the Discussion.

Figures 6, 7 and 8. Similar to the previous comment concerning the "immunological part", some inconsistencies between the transcriptomic data and the qPCR results were also observed. Moreover, in these cases, a significant influence of Tween 80 was also observed.

DISCUSSION

In general, I don't have major concerns to Discussion. Maybe only the Discussion is written in too general term. Sometimes lack is deeper analysis/discussion, for examples set of immune-related genes which expression changed in two tested diet regimes is very interesting and should be briefly discussed. Also analysis of transcriptomic data concerning metabolic changes are almost omitted in the Discussion.

Lines 316-318. I suggest to "softening" this sentence. Few articles have investigated the effects of larval diet on the immune response of adult individuals after *B. bassiana* infection.

Lines 322. It is better to say "hemocyte count" or "hemocyte number"

MATERIALS AND METHODS

Lines 436-446. Please add the information about number of individuals per box and amount of food in the box.

Lines 481-482. What was real number of biological repetitions per treatment? Whether three individuals were pooled in one sample or analysed separately as three samples?

Line 498. Please add the information about concentration of RNA used for cDNA synthesis.

Line 507. Whether the primer efficiencies entitle to use $2^{-\Delta\Delta C_t}$ method?

Lines 507-508. Please add the information how many individuals were pooled as a one biological sample.

Lines 536--537. In my opinion the number of biological samples is definitely too small for this kind of enzymatic assay.

Line 597. Please add the accession number.

Reviewer #2

(Remarks to the Author)

The objective of this study was to examine the impacts of nymphal diet on adult immunity function, body macronutrient content, and the related gene expression patterns in an omnivorous plant bug, *Adelphocoris suturalis*. Throughout their development, nymphs were raised on either herbivore (solely plant, NP) or omnivore diet (a mix of plant and aphid prey, NO). Upon reaching the adulthood, these insects with different diet histories were challenged with fungal infection (*Beauveria bassiana*) and various aspects of their immunological and metabolic parameters, as well as the transcriptomic profiles were quantified as the infection progressed. Results showed that adults raised on NO expressed greater resistance to fungal infection compared to those raised on NP, which was explained by improved immune functions. The expression profile of the immune-related genes also aligned with this diet effect. This study also found that the NO group exhibited higher number of up-regulated metabolism-related genes than the NP group, providing mechanistic insights that improved immune function in the NO group is possibly linked to increased metabolic capacity. Overall, I think this is a well thought-out study addressing a question that is insofar not much touched in the field of insect nutritional immunology. The experiments were appropriately executed and the manuscript was generally well written. Results are also interesting, which may provide valuable insights into how the quality of diet consumed during development modulates adult immunity. However, the results reported in this work suggest only correlational relationships between nymphal diet and adult immunity. This aspect makes me question whether this work is outstanding enough to make a significant advancement in the field and to have broader implications in the scientific community. I have also identified a number of issues, which undermines the overall quality of

this submission. First, in many cases, I found the references cited in this manuscript (especially in the introduction section) to be inappropriate or outdated. While reading this manuscript, I received strong impression that the authors were not fully aware of the key literatures related to 1) the impact of juvenile diet on adult fitness, 2) nutritional effects on insect immune function, and 3) the nutritional ecology of omnivory (see my detailed comments below). Second, in the current study, the adult bugs from two different nymphal diet groups were challenged with a live fungus to induce immune responses. However, this way of experimentally infecting insects does not allow the experimenters to distinguish the gene expression patterns induced by immune activation from those by other pathogen-induced responses (unrelated to host immune activation). I think the narrative of this work could have been strengthened if it also treated or injected insects some immune response elicitors, such as lipopolysaccharides LPS. Third, this study did not take into account the gender of adult insects when analyzing immune function and macronutrient content and their gene expressions. I think this is potentially problematic because males and females differ considerably in their ability to mount immune function and also in macronutrient composition in many insects.

Specific or minor comments:

Ln 1: not familiar with the term “immune storage”. What does this mean?

Ln 2: “in a mirid bug”

Ln 17: I think the study insect, “*Adelphocoris suturalis*”, should be mentioned earlier in the abstract (not in Ln 26).

Ln 32: “Metabolic”? Rephrase.

Ln 38: References 1 and 2 should be replaced with some key papers related to this topic. For example: Monaghan, P. (2008). Early growth conditions, phenotypic development and environmental change. *Philosophical Transactions of the Royal Society B: Biological Sciences*, 363, 1635–1645 -

Ln 40: Reference 4 is not about insect. Replace.

Ln 49: “Eevi et al” should be “Savola et al. “

Ln 55, Refs 11 and 12: Please replace these references with some relevant recent reviews, such as Cotter, S.C. and Al Shareefi, E., 2022. Nutritional ecology, infection and immune defence—Exploring the mechanisms. *Current Opinion in Insect Science*, 50, p.100862.

Ponton, F., Tan, Y.X., Forster, C.C., Austin, A.J., English, S., Cotter, S.C. and Wilson, K., 2023. The complex interactions between nutrition, immunity and infection in insects. *Journal of Experimental Biology*, 226(24), p.jeb245714.

Ln 57-74: This paragraph needs to be reduced by 50%.

Ln 64: Not sure what this sentence means.

Ln 79-80: Rewrite.

Ln 83-103: This last paragraph of the introduction needs to include the some key hypotheses or predictions tested in this study.

Ln 128-129: “To investigate the reasons behind how ... to pathogenic fungi, we conducted deep RNA-seq on ..”.

Ln 197-199: Rewrite.

Ln 213-214: “which led to a significant decrease”? Rewrite.

Ln 327-330: Are there any relevant references supporting this statement.

Ln 333, ref 28: Please cite the following seminal references here, instead of ref 28..

Moret, Y. and Schmid-Hempel, P., 2000. Survival for immunity: the price of immune system activation for bumblebee workers. *Science*, 290(5494), pp.1166-1168.

Ln 349-350: Add a relevant reference.

Ln 370-411: This is a long paragraph. It is better to divide it into two paragraphs..

Discussion: What is completely missed in the current manuscript is a discussion of why the NO diet was a better diet for adult immunity than the NP diet. There is a large body of literature reporting nutritional benefits of omnivorous diets in terms of improved protein or nitrogen balance in insects. I think the authors need to discuss about the possibility that improved protein balance on omnivorous diet might have driven insects to invest more protein resources into immunity.

Figures 3,4,5,6,7, 8: In the same plot, statistically significant differences among treatments were represented by different letters in one case and by asterisks in the other. I think this should be consistent.

Reviewer #3

(Remarks to the Author)

The manuscript by Li et al. explores the impact of nymphal diet on adult immunity in *Adelphocoris suturalis*. The authors conducted extensive RNAseq experiments to identify differentially expressed immune- and metabolism-related genes in adults fed an omnivorous diet as nymphs compared to those fed a plant-based diet as nymphs. The study demonstrates that an omnivorous diet enhances immune responses and metabolism, leading to higher survival rates following infection with *Beauveria bassiana*. While the study is both interesting and significant, it requires substantial revisions before it can be considered for publication.

Structure, Style, and Clarity

The manuscript would benefit from a thorough revision to improve language, style, and structure. Below are specific suggestions for improvement:

- Abstract:

- o The abstract does not clarify the organism or organismal group being studied until the very end. To enhance clarity, I recommend stating at the beginning of the abstract that the study focuses on insect immunity.

- Introduction:

- o The introduction would benefit from a more detailed discussion of constitutive and induced immune responses in insects. This is particularly important in the context of the parameters measured in this study (PO activity, PPO activity, and hemocyte load). For instance, how do these parameters in particular contribute to insect immunity, and how are they regulated? Adding this context will help readers understand the significance of the findings.

- Figures:

- o The figures generally enhance the manuscript's comprehensiveness. However, in Figure 3b, the shown pathway looks very out-of-place and I initially thought it was put there by mistake. Please improve the presentation to clarify that this pathway represents the genes validated by qPCR. This issue is handled much better in Figures 6b, 7b, and 8b.

- o For the heatmaps, please explicitly indicate what the colors represent (e.g., fold change) in both the legend and the figure captions.

Methods

The methods are described in good detail overall, but some sections lack critical information and clarity. Below are specific areas that need attention:

- Identification of Immune-Related Genes:

- o The authors mention using tBLASTn searches to identify immune-related genes (Line 161). However, the description is incomplete. Please provide details on the query and database used, the search parameters, and how the immune-related genes were identified.

- Selection of Genes for RT-qPCR Validation:

- o The manuscript briefly mentions the validation of immune-related genes (Lines 179–182) and macro-nutrient metabolism pathway genes (Lines 256–271) via RT-qPCR. Please elaborate on how these genes were selected and the rationale behind their selection.

- Transcriptome Assembly Quality:

- o The authors de-novo assembled a transcriptome, but I cannot find anything about the quality of the assembly. In the methods, the authors state that they used BUSCO (Line 491: for retrieving transcript ORF? Please clarify this). Please provide information on the software version, reference database, and quality metrics (e.g., completeness scores). Additionally, software versions should be included throughout the methods section for consistency.

Discussion

The discussion would benefit from a deeper exploration of the mechanisms underlying the observed results. Specifically:

- The authors should discuss how the upregulation of immune-related genes in the omnivorous diet (NO) group enhances survival following *B. bassiana* infection. More information on the infection route of *B. bassiana* and the roles of PO, PPO, and hemocytes in combating its spores would strengthen both the introduction and the discussion. This would provide important context for understanding the observed immune responses.

Minor Comments

- Line 50: Please explain the abbreviation "P:C."

- Line 141: Clarify what is meant by "-log10 values."

- Figure 2: The font size is too small and should be increased for better readability.

- Figure 2c, e: The color codes in panels c and e are inconsistent. Please standardize the color codes for easier comparison. Additionally, using the same x-axis scale for both plots would improve visual comparability.

Version 1:

Reviewer comments:

Reviewer #1

(Remarks to the Author)

Thank you very much for the opportunity to review the revised version of the manuscript titled "Nymphal diets boost adults' immunity via strengthened constitutive immunity and metabolic capacity in a mirid bug." The authors have addressed all of my suggestions, mostly in a very satisfactory manner, including the performance of new experiments. The additional information added to the main text has led to better clarification of the methodology used and improved data presentation. I also appreciate that the authors thoroughly revised the Discussion section and provided a more specific explanation of the observed physiological changes in the mirid bug following different dietary treatments. However, in my opinion, the current version of the Discussion is too long, as the authors have added approximately nine pages of new information and explanations. I believe this section could be shortened, and the authors should focus on extracting and presenting the most important points.

Reviewer #2

(Remarks to the Author)

I have carefully read the responses of the authors to my queries earlier. I think most of my comments are meticulously considered by the authors. The addition of a new experiment testing the impacts of immune elcitor greatly strengthen the conclusion of this work. I am generally positive about this revision and thus have no further comments on this work.

Reviewer #3

(Remarks to the Author)

I appreciate the authors' detailed responses and think they significantly revised and improved the manuscript. However, I still have a few minor issues that I recommend addressing.

Comment 6:

The authors have added a more detailed description of the immune gene identification in the Methods section, but the general methodology should also be apparent from simply reading the results. However, in Line 170, it only says "Using tBLASTn searches, we identified 695 immune-related unigenes", which does not provide enough clarity for readers. Please rephrase this as follows: "Blasting a reference set of insect immune-related genes against the *A. suturalis* genome, we identified ..."

Comment 8:

The authors have clarified the use of BUSCO and added the missing information about software versions, reference database and some quality metrics. However, I would appreciate if they gave the complete BUSCO result, including duplicated BUSCOs, since this is important to evaluate the quality of the transcriptome.

Additional minor points:

- Please make sure to use the terms "nymph" and "larva" correctly for hemimetabolous and holometabolous insects, respectively (line 46, for example).
- Please change the term "nymph diet" to "nymphal diet" throughout the manuscript
- Line 76-77: rephrase: "... that can cause serious damage to various crops including cotton, soybean, maize and fruit trees"

Reviewer #1 (Remarks to the Author):

Comment 1: The manuscript entitled “Nymphal diets boost adults’ immunity via immune storage and enhanced metabolism in a Mirids bug” present interesting, new data concerning how the nymphal diet may affect the functioning of immune system of adult individuals. Generally, the manuscript is well written but some of the issues concerning research methodology and results should be clarified by Authors. For this reason, I recommend major revision. Detailed comments you can find below.

Response: We sincerely thank the reviewer for recognizing the novelty and significance of our study, and we also greatly appreciate the constructive feedback on our manuscript, which helped improve the clarity and rigor of our manuscript. Followed your suggestions, we have carefully revised the manuscript to clarify methodological details, refine the Introduction and Results presentation, and standardize figure presentation and statistical annotations. We have also thoroughly checked all data of the transcriptomic and qPCR data to confirm accuracy and substantially expanded the Discussion to provide detailed mechanistic analyses of how early-life omnivory enhances adult immune responses.

Detailed point-by-point responses are provided below.

INTRODUCTION

Comment 2: Lines 57-74. I understand the Authors intention concerning division of immune system mechanisms into constitutive and induced immunity. However, if the article is directed to broader audience, Authors should also explain reader that insect immunity based mostly on the innate mechanisms etc.

Response: Thank you for the insightful suggestion. we have revised the relevant section to explicitly state that insects lack adaptive immunity and instead rely primarily on two major forms of innate immunity—constitutive and induced—to combat pathogens (**Lines 53-55**).

Comment 3: Line 72. Some articles suggest that there is more than one lysozyme. Also, in several articles, lysozymes are wrongly classified as AMPs.

Response: Thank you for this valuable comment, which helped improve the accuracy of our Introduction. In the revised Introduction (**Lines 73-74**), we have revised the classification by distinguishing lysozymes from AMPs, and used the plural form to reflect that insects have more than one lysozyme.

RESULTS

Comment 4: Lines 107-120. Why the survival experiments lasted only 8 days?

Response: Thank you for this valuable comment, which helped clarify our experimental design. In our preliminary experiments, we observed that survival rates of *B. bassiana*-infected adults declined rapidly between 4-8 days post-infection and showed no significant changes thereafter (see figure below). Based on this, we selected an 8-day monitoring period to capture the critical window of pathogen-induced survival differences while minimizing potential confounding from age-related or background mortality.

Comment 5: Line 115 and Line 116. Please add the SEM or SD value for the LT50.

Response: As suggested, we have added the SEM values for LT50 in the revised manuscript (**Lines 128**).

Comment 6: Lines 121-125. The results related to the differences between diets may be separate from the results of the survival experiment. I think it is good to start with this information. Additionally, please explain why the lipid content and the fatty acid composition were not analysed? On the basis of the presented results, the Authors are aware of the role of lipids in energy metabolism

and their direct and indirect roles in shaping insect immune system activity.

Response: Thank you for this helpful suggestion. In the revised manuscript, we have restructured this section so that the results on the macronutrient composition of the dietary ingredients precede the survival analysis (**Lines 108-117**), as recommended. We have also newly quantified triglyceride content for both dietary ingredients and updated the corresponding Results, Figure, and Legend.

We did not analyze fatty acid composition, because, in insects, most fatty acids are stored as triglycerides (Wolins, Brasaemle et al. 2006, Siddiqui, Zeiri et al. 2025), which can be rapidly mobilized via lipolysis during energy-demanding processes such as immune activation (Bland 2023). Free fatty acids generally occur at low concentrations and provide limited information on total lipid reserves (Kaczmarek and Bogus 2021). Thus, triglyceride content is widely used as a proxy for lipid storage in insect physiological studies, and we have followed this convention here.

Reference:

Siddiqui, S. A., Zeiri, A. & Asif Shah, M. Insect Lipids as Novel Source for Future Applications: Chemical Composition and Industry Applications-A Comprehensive Review. *Food Sci Nutr* 13, e70553, doi:10.1002/fsn3.70553 (2025).

Wolins, N. E., Brasaemle, D. L. & Bickel, P. E. A proposed model of fat packaging by exchangeable lipid droplet proteins. *FEBS Lett.* 580, 5484-5491, doi:10.1016/j.febslet.2006.08.040 (2006).

Bland, M. L. Regulating metabolism to shape immune function: Lessons from *Drosophila*. *Seminars in Cell & Developmental Biology* 138, 128-141, doi:10.1016/j.semcdb.2022.04.002 (2023).

Kaczmarek, A. & Bogus, M. The metabolism and role of free fatty acids in key physiological processes in insects of medical, veterinary and forensic importance. *PeerJ* 9, e12563, doi:10.7717/peerj.12563 (2021).

Revised Figure 1

Comment 7: Fig 1C and D. The abbreviation of group should be consistent in whole manuscript. I suggest change the group names on the NO and NP.

Response: Thank you for the valuable suggestion. We fully agree on the importance of consistent labeling throughout the manuscript. Upon careful review, we noted that Figures 1c and 1d (now revised as Figures 1b and 1c) present the nutrient composition of the dietary ingredients (mung bean sprouts and aphids) used in the nymphal diet treatments, rather than data from the experimental groups themselves. Since NO (nymphs fed on an omnivorous diet) and NP (nymphs fed on a phytophagous diet) denote combinations of these ingredients provided to nymphs, labeling the raw ingredients as NO/NP may cause confusion. We therefore retained the labels “mung bean sprouts” and “aphids” in Figures 1b and 1c, while using “NO” and “NP” consistently in all subsequent figures that compare experimental groups.

Comment 8: Figures 1, 3, 4, 5, 6, 7 and 8. I think the plot style should be unified, including indication of statistically significant changes. In the current version, data are sometimes presented as bar plots, sometimes as scatter plots with bars, and sometimes as scatter plots. Additionally, sometimes in one graph, we see different styles of indication of statistically significant changes. Authors should choose whether the significant changes will be indicated as asterisks or different

letters.

Response: Thank you for this helpful suggestion. We have unified the presentation across all relevant figures (Figures 1, 3-9; original Figures 5-8 are now Figures 6-9 due to the insertion of a new Figure 5 in the revised manuscript) by using bar plots with individual data points to consistently represent data.

Regarding the notation of statistical significance, we now use only letters in Figures 3-9 to ensure a consistent annotation style. Given that our analyses involved two-way ANOVA, we used the following scheme to clarify the sources of significance: “X, Y” denote diet effects; “ α , β ” indicate infection effects; and “a, b, c, d” represent significant differences among all treatment combinations when a significant interaction effect was detected. Corresponding figure legends have been updated accordingly for clarity.

Comment 9: Lines 131-133 and Lines 471-476. Please explain why this three time-points were chosen?

Response: Thank you for raising this important point. Our sampling strategy was designed to capture immune and physiological changes across distinct stages of pathogen challenge: 0 dpi represents the baseline before infection; 2 dpi, corresponding to the early infection phase (~10% mortality in the NP-Bb group); and 6 dpi, representing the mid-to-late stage of infection progression (~50% mortality in the NP-Bb group). We have now clarified this rationale in both the Results section (**Lines 139–142**) and the Methods section (**Lines 671-676**).

Comment 10: Lines 169-199 and Figure 3. A comparison of the transcriptomic analysis results, which are presented as heatmaps, and the qPCR results raises some doubts. On the basis of the heatmap, we may assume that the use of Tween 80 as a control may substantially affect the expression levels of immune-related genes even two days after its application (compare NO vs NO-Ctrl and NP vs NP-Ctrl). In addition, most of the qPCR analyses did not confirm the results of the transcriptomic data. I know that this is normal that may exist the differences between transcriptomic data and qPCR results which is related to the methodology of these techniques. However, in the case of some genes, the trends between the control individuals (NO-Ctrl and NP-Ctrl) and insects treated with *B. bassiana* were opposite, for example, in the case of Lys on day 6 (NP groups), in the case of

Toll4b on day 2, or in the case of ALo2a on day 2 (NP groups). There are many such examples, so please carefully revised your data.

Response: We appreciate the reviewer's careful examination of our data and the opportunity to clarify these points.

1. On the potential effects of Tween 80 treatment:

At 0 dpi (pre-infection baseline), adults from both NP and NO treatment also underwent the same Tween 80 immersion procedure. We have clarified this in the Methods section (*0 days post-infection (dpi), representing the pre-infection baseline, when adults from both dietary treatments had undergone the same Tween 80 treatment. Lines 671-676*) and revised the figure 3a labels at 0 dpi from “NO”/“NP” to “NO-Ctrl”/“NP-Ctrl”, consistent with our original definition of “Ctrl” as adults treated with Tween 80. To assess whether Tween 80 affects immune gene expression, we compared representative immune-related genes between untreated adults and Tween 80–treated adults, and no gene showed significant differences (see figure below). These results confirm that Tween 80 treatment did not influence immune gene expression. We acknowledge that the marked differences between control groups at 0 dpi and 2 dpi may have contributed to the reviewer’s concern; however, these differences are unlikely to result from Tween 80 exposure and more likely reflect hormonally regulated prophylactic immunity during molting, a stage when the soft new cuticle is vulnerable to microbial invasion or mechanical damage—consistent with reports in newly emerged *Ae. aegypti* (Zhang, Dong et al. 2017) and *D. melanogaster* (An, Dong et al. 2012) adults. This interpretation is also discussed in the revised Discussion (**Lines586-592**)

2. On non-concordant between qPCR and transcriptomic results:

We have carefully rechecked all data and confirm that the values presented are accurate. As noted by the reviewer, methodological differences between RNA-seq and qPCR can lead to partial inconsistencies, a phenomenon well-documented by Everaert et al. (2017), who also reported that opposite trends between the two methods can occasionally occur. In our dataset, most cases cited by the reviewer (e.g., *Lys* at 6 dpi in NP; *Alo2a* at 2 dpi in NP; *Toll4b* at 2 dpi) show opposite trends between RNA-seq and qPCR; however, both methods consistently indicated no statistically significant differences between treatments. Statistically significant opposite trends (*Pelle1* at 2 dpi NP-Bb vs NP-Ctrl; *SPZ* at 2 dpi NO-Bb vs NP-Bb; *cSPI6* at 6 dpi NP-Bb vs NP-Ctrl and NO-Bb vs NO-Ctrl; *Tube2* at 6 dpi NO-Bb vs NO-Ctrl) occurred in only accounting for only 7.4% of all

comparisons. These rare cases do not affect the main conclusions of our study, and we appreciate the reviewer's attention to this important point.

Reference:

Zhang, H. *et al.* Relish2 mediates bursicon homodimer-induced prophylactic immunity in the mosquito *Aedes aegypti*. *Sci Rep* 7, doi:10.1038/srep43163 (2017).

An, S. *et al.* Insect neuropeptide bursicon homodimers induce innate immune and stress genes during molting by activating the NF- κ B transcription factor Relish. *PLoS One* 7, e34510, doi:10.1371/journal.pone.0034510 (2012).

Everaert, C., Luybaert, M., Maag, J.L.V. *et al.* Benchmarking of RNA-sequencing analysis workflows using whole-transcriptome RT-qPCR expression data. *Sci Rep* 7, 1559 (2017). <https://doi.org/10.1038/s41598-017-01617-3>

Comment 11: Lines 175-178. This part better fits the Discussion than the Results.

Response: We agree. This sentence has been moved to the Discussion (**Lines 582-592**).

Comment 12: Line 204. Should be “hemocyte number” or “hemocyte count”.

Response: Corrected. We have replaced “hemocyte concentrations” with “hemocyte counts” at **Line 213** and throughout the manuscript to ensure consistency.

Comment 13: Lines 215-216. This sentence should be moved to the Discussion.

Response: Thanks. This sentence has been relocated to the Discussion section. (**Lines 367-370**).

Comment 14: Figures 6, 7 and 8. Similar to the previous comment concerning the “immunological part”, some inconsistencies between the transcriptomic data and the qPCR results were also observed. Moreover, in these cases, a significant influence of Tween 80 was also observed.

Response: We appreciate the reviewer’s careful examination of our data. Figures 6, 7, and 8 (Figure 7, 8, 9 in revised manuscript respectively) present the expression patterns of carbohydrate-, lipid-, and amino acid metabolism-related genes, which in our results showed a positive correlation with immune responses. The issues noted here regarding Tween 80 effects and RNA-seq/qPCR inconsistencies follow the same reasoning as addressed in **Comment 10**. Briefly:

(i) **Tween 80:** adults from both diet groups at 0 dpi underwent identical Tween 80 treatment, which we have clarified in the Methods section (**Lines 671-673**). The labels at 0 dpi in Figures 7a/8a/9a (prior submission: Figs. 6a/7a/8a) were revised from “NO”/“NP” to “NO-Ctrl”/“NP-Ctrl” to avoid confusion. Immune-related gene validation (see supplementary data in Comment 10) showed no effect of Tween 80, and the positive correlation between metabolic- and immune-related gene expression patterns suggests that the transient upregulation of metabolic genes at 0 dpi may be a normal physiological phenomenon rather than an effect of Tween 80 treatment. Because immune gene validation demonstrated no Tween 80 effect and our study focuses on immunity, we did not perform additional validation for metabolic genes. Instead, we interpret their transient 0 dpi upregulation as a normal physiological phenomenon linked to immune capacity rather than an effect of Tween 80.

(ii) **RNA-seq vs qPCR results:** We have carefully rechecked all data and confirm that the values presented in the manuscript are accurate. As in Comment 10, differences between RNA-seq and qPCR are within the expected methodological variation, with statistically significant opposite trends occurring in only 4.44%, 6.67%, and 3.17% of cases in Figures 7, 8, and 9, respectively. We believe

that these rare cases do not affect the main conclusions of our study.

DISCUSSION

Comment 15: In general, I don't have major concerns to Discussion. Maybe only the Discussion is written in too general term. Sometimes lack is deeper analysis/discussion, for examples set of immune-related genes which expression changed in two tested diet regimes is very interesting and should be briefly discussed. Also analysis of transcriptomic data concerning metabolic changes are almost omitted in the Discussion.

Response: We appreciate the reviewer's valuable suggestion. In the revised Discussion (**Lines 375–508**), we have substantially expanded the mechanistic interpretation of our findings, incorporating deeper analyses of early-life diet-induced changes in hemocyte counts, PPO/PO activity, and immune- and metabolism-related gene expression.

Specifically:

- 1. We discuss the potential mechanisms by which early-life omnivory strengthened constitutive immune traits in adults.** This conclusion is supported by our observation that NO adults exhibited elevated baseline levels of PPO activity and hemocyte counts under uninfected conditions—hallmark indicators of constitutive immunity. We propose that this enhancement likely reflects greater developmental investment into immune function, potentially driven by higher protein intake during the nymphal stage (**Lines 375–395** for details).
- 2. We analyze how early-life omnivory durably enhanced adults' inducible immune responses, despite both treatment groups sharing an identical phytophagous diet during adulthood,** and explore three non-mutually exclusive mechanisms:

First, omnivorous juveniles accumulated more protein—a critical resource for immune tissue development, PPO production, and AMP synthesis. Notably, this nutritional advantage diminished over time as all adults transitioned to a uniform phytophagous diet, indicating that mechanisms beyond nutrient accumulation contributed to the sustained immune enhancement (**Lines 408–422** for details).

Second, early omnivory promoted structural investment into immune systems, as reflected by elevated hemocyte counts at emergence, which likely supported hemocyte proliferation (**Lines**

423-433 for details) and immune effector production (**Lines 434-454** for details) during infection.

Third, early dietary experience appeared to establish a more robust metabolic infrastructure, capable of meeting the energetic and biosynthetic demands of inducible immunity (**Lines 473-508** for details).

- 3. We highlight that the immune-related genes upregulated in the NO group were predominantly enriched in the fungal-responsive Toll signaling pathway and its downstream antifungal effectors**, including *Alo* and *lysozymes*, indicating targeted reinforcement of antifungal defenses. (**Lines 458-462** for details).
- 4. We have elaborated on the transcriptomic evidence for diet- and infection-induced metabolic changes**, detailing upregulated genes in lipid catabolism, glycolysis, the TCA cycle, and amino acid metabolism, and discuss how these metabolic pathways contribute to immune activation by supplying both energy and biosynthetic precursors (**Lines 473-508** for details).

We believe these additions address the reviewer's concerns and provide a clearer, more comprehensive mechanistic framework linking early dietary experience to adult immunity.

Comment 16: Lines 316-318. I suggest to “softening” this sentence. Few articles have investigated the effects of larval diet on the immune response of adult individuals after *B. bassiana* infection.

Response: Thank you for this helpful suggestion. We acknowledge that the original phrasing (“first examined”) could be misinterpreted as implying novelty, which was not our intent. Our aim was to indicate that this assessment represented the initial step of our experimental design. To clarify, we have revised the sentence as follows:

“These complexities highlight the limitations of assessing immune capacity based on single traits. To reduce such bias, we began by assessing survival following *B. bassiana* infection as a comprehensive phenotypic indicator of adult immune resistance.” (**Line 359-362**)

Comment 17: Lines 322. It is better to say “hemocyte count” or “hemocyte number”

Response: Revised as suggested. We have replaced “hemocyte load” with “hemocyte count” at **Line 364** and throughout the manuscript to ensure consistency.

MATERIALS AND METHODS

Comment 18: Lines 436-446. Please add the information about number of individuals per box and amount of food in the box.

Response: Revised as suggested, we have included this information in the revised manuscript (**Lines 636-638**): “Nymphs and adults were reared in plastic cages (225 mm long, 150 mm wide, 110 mm high) at a density of approximately 60 individuals per cage. Nymphs were fed fresh mungbean sprouts, and adults were fed fresh green beans. All food was supplied daily in sufficient quantities. In addition, aphids (*Acyrtosiphon pisum*) were provided daily in sufficient amounts as a dietary supplement for both nymphs and adults.”

Comment 19: Lines 481-482. What was real number of biological repetitions per treatment? Whether three individuals were pooled in one sample or analysed separately as three samples?

Response: Thank you for raising this point. We have clarified in the revised Methods section (**Lines 681-684**) that three biological replicates were prepared for each treatment and time point, with each replicate consisting of three individuals pooled, homogenized, and processed as a single sample for RNA extraction and subsequent analysis.

Comment 20: Line 498. Please add the information about concentration of RNA used for cDNA synthesis.

Response: Thank you for pointing this out. In the revised manuscript (**Line 739**), we have specified that 1 µg of total RNA was used for cDNA synthesis in each sample, regardless of its individual concentration. This corrects the original unit error (“1 µL”) in the submitted version.

Comment 21: Line 507. Whether the primer efficiencies entitle to use $2^{-\Delta\Delta C_t}$ method?

Response: Thank you. All primers were validated to have amplification efficiencies between 90% and 110%, which meets the accepted criteria for applying the $2^{-\Delta\Delta C_t}$ method. This information has been added to the revised manuscript (**Lines 751-753**) and is detailed in **Supplementary Table 1**.

Comment 22: Lines 507-508. Please add the information how many individuals were pooled as a

one biological sample.

Response: Thank you for your comment. We have specified in the revised manuscript that each biological replicate consisted of three pooled individuals (**Lines 747-749**).

Comment 23: Lines 536--537. In my opinion the number of biological samples is definitely too small for this kind of enzymatic assay.

Response: We agree with the reviewer. In the revised manuscript, we have increased the number of biological replicates for the PPO and PO activity assays to six. The updated results are highly consistent with our original findings. We have updated the Methods (**Lines 783-784**), statistical analyses, and Figure 4 accordingly.

Comment 24: Line 597. Please add the accession number.

Response: Thank you. We have added the accession number [PRJNA1191641] in the revised manuscript (**Line 864-865**).

Reviewer #2 (Remarks to the Author):

Comment 1: The objective of this study was to examine the impacts of nymphal diet on adult immunity function, body macronutrient content, and the related gene expression patterns in an omnivorous plant bug, *Adelphocoris suturalis*. Throughout their development, nymphs were raised on either herbivore (solely plant, NP) or omnivore diet (a mix of plant and aphid prey, NO). Upon reaching the adulthood, these insects with different diet histories were challenged with fungal infection (*Beauveria bassiana*) and various aspects of their immunological and metabolic parameters, as well as the transcriptomic profiles were quantified as the infection progressed. Results showed that adults raised on NO expressed greater resistance to fungal infection compared to those raised on NP, which was explained by improved immune functions. The expression profile of the immune-

related genes also aligned with this diet effect. This study also found that the NO group exhibited higher number of up-regulated metabolism-related genes than the NP group, providing mechanistic insights that improved immune function in the NO group is possibly linked to increased metabolic capacity. Overall, I think this is a well thought-out study addressing a question that is insofar not much touched in the field of insect nutritional immunology. The experiments were appropriately executed and the manuscript was generally well written. Results are also interesting, which may provide valuable insights into how the quality of diet consumed during development modulates adult immunity.

Response: We sincerely thank the reviewer for the comprehensive summary, positive evaluation, and constructive feedback on our manuscript. We are encouraged that you found our study well designed, interesting, and relevant to advancing the field of insect nutritional immunology. Your thoughtful suggestions have been very helpful in improving the clarity and rigor of our work. We have carefully revised the manuscript by updating key references, adding immune elicitation experiments, clarifying methodological details, refining the content of Introduction, restructured the Discussion to explain why the NO diet enhances adult immunity, and standardized statistical notation across all figures. Detailed point-by-point responses are provided below.

Comment 2: However, the results reported in this work suggest only correlational relationships between nymphal diet and adult immunity. This aspect makes me question whether this work is outstanding enough to make a significant advancement in the field and to have broader implications in the scientific community.

Response: We appreciate the reviewer's thoughtful comment, which helped us better define the scope and significance of our current findings.

We fully acknowledge that our study establishes a strong correlation between nymphal diet and adult immune performance but does not yet provide direct causal evidence through mechanistic interventions. While we did not experimentally manipulate specific immune or metabolic pathways, our findings are supported by convergent evidence across multiple biological levels—including phenotypic (survival), physiological (hemocyte count, PO/PPO activity and macronutrient profiles), and molecular (qPCR and transcriptomic) measures. This consistency across levels strengthens the inference of a biologically meaningful link between early-life diet and adult immunity, likely shaped

through developmental physiological and molecular processes. In the revised Discussion (**Lines 611-617**), we have explicitly noted this limitation and outlined future directions—such as RNAi-mediated knockdown of key genes and nutrient-specific manipulation experiments—to establish causality and elucidate underlying mechanisms.

Comment 3: I have also identified a number of issues, which undermines the overall quality of this submission. First, in many cases, I found the references cited in this manuscript (especially in the introduction section) to be inappropriate or outdated. While reading this manuscript, I received strong impression that the authors were not fully aware of the key literatures related to 1) the impact of juvenile diet on adult fitness, 2) nutritional effects on insect immune function, and 3) the nutritional ecology of omnivory (see my detailed comments below).

Response: We appreciate the reviewer's insightful comment on the appropriateness and currency of the cited literature. We have carefully addressed all points by replacing outdated or less relevant references with more appropriate and recent sources, including the reviewer-recommended papers, thereby improving the clarity, relevance, and scientific rigor of the manuscript. Detailed revisions are provided under Comments 10, 11, 13, 21, 22, and 23 in our point-by-point responses.

Comment 4: Second, in the current study, the adult bugs from two different nymphal diet groups were challenged with a live fungus to induce immune responses. However, this way of experimentally infecting insects does not allow the experimenters to distinguish the gene expression patterns induced by immune activation from those by other pathogen-induced responses (unrelated to host immune activation). I think the narrative of this work could have been strengthened if it also treated or injected insects some immune response elicitors, such as lipopolysaccharides LPS.

Response: Thank you for this valuable suggestion, which prompted us to strengthen the mechanistic basis of our conclusions. To directly assess whether the enhanced immunity in NO adults reflects a greater capacity for immune activation rather than non-immune pathogen-induced effects, we conducted a supplementary experiment by injecting β -1,3-glucan—a fungal-specific immune elicitor—into 1.5-days-old adults from both diet groups. This timing was selected to align both the developmental stage and immune activation window with those of the *B. bassiana*-infected group, given that *B. bassiana* typically penetrates the host cuticle within 24-48 h (Lai, Chen et al. 2017,

Wang, Lai et al. 2023). Samples were collected at 6 h, 12 h (developmentally corresponding to 2 dpi in the *B. bassiana* infection group), 24 h, and 108 h (corresponding to 6 dpi) post-injection. We measured PO and PPO activities, hemocyte counts, and the expression of representative immune-related genes (consistent with those used for qPCR validation in the *B. bassiana* infection assay). The results showed that β -1,3-glucan induced robust immune activation in both diet groups, but the NO group consistently exhibited stronger responses than NP across all parameters. These results closely paralleled the patterns observed under *B. bassiana* infection, supporting the interpretation that the enhanced immunity in the NO group reflects a greater capacity for immune activation rather than non-immune pathogen-induced effects.

We have incorporated these new experiments into the revised manuscript, updating the Methods (**Lines 785-804**), Results (**Lines 226-260**), figures (Figure 5 in revised manuscript), and figure legends accordingly.

Reference:

Wang, L. et al. The ASH1-PEX16 regulatory pathway controls peroxisome biogenesis for appressorium-mediated insect infection by a fungal pathogen. *Proc. Natl. Acad. Sci.* 120, e2217145120, doi:10.1073/pnas.2217145120 (2023).

Lai, Y. et al. In vivo gene expression profiling of the entomopathogenic fungus *Beauveria bassiana* elucidates its infection stratagems in *Anopheles mosquito*. *Sci China Life Sci* 60, 839-851, doi:10.1007/s11427-017-9101-3 (2017).

Figure 5 in the revised manuscript

Comment 5: Third, this study did not take into account the gender of adult insects when analyzing immune function and macronutrient content and their gene expressions. I think this is potentially problematic because males and females differ considerably in their ability to mount immune function and also in macronutrient composition in many insects.

Response: Thank you for raising this important point. We apologize for not clearly specifying this information in the original manuscript. All insects used in this study were females, thereby eliminating potential confounding effects of sex-specific variation in immune or metabolic traits. This information has been added to the Methods section (**Lines 642**).

Comment 6: Ln 1: not familiar with the term “immune storage”. What does this mean?

Response: Thank you for pointing this out. We agree that “immune storage” is not a standard term in insect immunology and may cause confusion. In the revised manuscript, we have replaced it with the more widely used term “constitutive immunity” (**Line 2**), which refers to baseline immune traits present in the absence of infection—such as the prophenoloxidase (PPO) activity and circulating hemocyte counts we measured in adults under uninfected conditions. These traits represent the foundational immune capacity established during nymphal development, which, in our findings, may also underpins the enhanced inducible immune responses observed after pathogen challenge.

Comment 7: Ln 2: “in a mirid bug”

Response: Corrected as suggested. We have replaced “in a Mirids bug” with “in a mirid bug” at **Line 2**

Comment 8: Ln 17: I think the study insect, “*Adelphocoris suturalis*”, should be mentioned earlier in the abstract (not in Ln 26).

Response: Thank you for the suggestion. We have moved *Adelphocoris suturalis* earlier in the abstract (now in **Line 14**)

Comment 9: Ln 32: “Metabolic”? Rephrase.

Response: Corrected. We have replaced “Metabolic” with “Metabolism” at **Line 28**

Comment 10: Ln 38: References 1 and 2 should be replaced with some key papers related to this topic. For example:

Monaghan, P. (2008). Early growth conditions, phenotypic development and environmental change. *Philosophical Transactions of the Royal Society B: Biological Sciences*, 363, 1635–1645

Response: Revised accordingly. We have replaced the previous Refs 1 and 2 with Monaghan (2008) and Carvajal-Lago, Ruiz-Lopez et al (2021)(**Lines 34**)

Reference:

Carvajal-Lago, L., Ruiz-Lopez, M. J., Figuerola, J. & Martinez-de la Puente, J. Implications of diet

on mosquito life history traits and pathogen transmission. *Environ. Res.* 195, 110893, doi:10.1016/j.envres.2021.110893 (2021).

Comment 11: Ln 40: Reference 4 is not about insect. Replace.

Response: Revised accordingly. We have replaced the previous Ref 4 with a relevant insect study on honeybees (Nicholls et al., 2021) at **Line 36**

Reference:

Nicholls, E., Rossi, M. & Niven, J. E. Larval nutrition impacts survival to adulthood, body size and the allometric scaling of metabolic rate in adult honeybees. *J. Exp. Biol.* 224, doi:10.1242/jeb.242393 (2021)

Comment 12: Ln 49: “Eevi et al” should be “Savola et al. “

Response: Corrected. We have replaced “Eevi” with “Savola” at **Line 45**

Comment 13: Ln 55, Refs 11 and 12: Please replace these references with some relevant recent reviews, such as

Cotter, S.C. and Al Shareefi, E., 2022. Nutritional ecology, infection and immune defence—Exploring the mechanisms. *Current Opinion in Insect Science*, 50, p.100862.

Ponton, F., Tan, Y.X., Forster, C.C., Austin, A.J., English, S., Cotter, S.C. and Wilson, K., 2023. The complex interactions between nutrition, immunity and infection in insects. *Journal of Experimental Biology*, 226(24), p.jeb245714.

Response: Revised as suggested. Refs 11 and 12 have been replaced with Cotter & Al Shareefi (2022) and Ponton et al. (2023) (**Line 51**).

Comment 14: Ln 57-74: This paragraph needs to be reduced by 50%.

Response: Thank you for this helpful suggestion. We have substantially revised this paragraph by eliminating redundancies, tightening the language, and reorganizing the content, resulting in roughly a 50% reduction in the original descriptive content.

In addition, in response to comments from another reviewer, we inserted brief clarifications of the immune mechanisms associated with the functional parameters assessed in our study (e.g.,

PPO/PO activity, hemocyte count, AMP expression) to aid interpretation by a broader readership, As a result, while the total word count of the paragraph is similar to the original, the content is now more concise, focused, and directly relevant to the study objectives (**Line 53-74**).

Comment 15: Ln 64: Not sure what this sentence means.

Response: Thank you for pointing this out. We agree that the original sentence (“Induced immunity is always coupled with this line to fight infection.”) was unclear. We have removed this sentence and revised the surrounding text to improve clarity and logical flow (**Lines 58**).

Comment 16: Ln 79-80: Rewrite.

Response: Thank you for the suggestion. We have revised the sentence for clarity. The updated version reads: “This omnivorous feeding strategy allows for flexible manipulation of juvenile dietary inputs by combining plant- and animal-derived components with distinct nutrient compositions.” (**Line 79-82**)

Comment 17: Ln 83-103: This last paragraph of the introduction needs to include the some key hypotheses or predictions tested in this study.

Response: Thank you for this helpful suggestion. We have revised the final paragraph of the Introduction to explicitly state the core hypothesis and related predictions. Specifically, based on the observed phenotypic difference that adults reared on an omnivorous nymphal diet (NO) exhibited significantly higher survival than those reared on a phytophagous diet (NP) following *Beauveria bassiana* infection, **we hypothesized that this survival advantage may reflect diet-induced enhancement of immune capacity**. To test this, we employed RNA-seq to compare gene expression profiles across dietary treatments at early and late infection stages, and assessed immune function by quantifying PO/PPO activity, hemocyte counts, and immune-related gene expression. Finally, we measured macronutrient content and the expression of associated metabolic genes to examine **whether immune performance was linked to dietary regulation of host macronutrient reserves and associated metabolic processes** (**Lines 93-106**).

Comment 18: Ln 128-129: “To investigate the reasons behind how ... to pathogenic fungi, we

conducted deep RNA-seq on ..”.

Response: Thank you. The punctuation has been corrected (**Lines 136-137**).

Comment 19: Ln 197-199: Rewrite.

Response: Thank you for the helpful suggestion. We have replaced the original sentence with the following clearer and more precise description: “At 6 dpi, a significant interaction between the nymphal diet and infection was detected exclusively for *SPZ*, whose transcriptional level peaked in the NO-Bb group (Fig 3b; Supplementary Table 3a). The transcriptional levels of *GNBP1e* and *Tube2* was primarily driven by infection: *GNBP1e* was significantly upregulated, whereas *Tube2* was downregulated. By comparison, the transcriptional levels of *Toll4b* and *Lys* were mainly influenced by diet, with significantly higher transcript abundance in the NO group than in the NP group. Notably, *cSPI6* was independently regulated by both infection and diet, showing infection-induced upregulation and higher expression in the NO group.” (**Lines 201-210**).

Comment 20: Ln 213-214: “which led to a significant decrease”? Rewrite.

Response: Thank you for pointing this out. The sentence has been revised to: “In contrast, PPO activity declined significantly in both diet groups compared to their respective controls.” (**Lines 223-225**).

Comment 21: Ln 327-330: Are there any relevant references supporting this statement.

Response: Thank you. We have added two relevant references (Zhang et al., 2017; An et al., 2012) to support this statement (**Lines 591-592**).

Reference:

Zhang, H. et al. Relish2 mediates bursicon homodimer-induced prophylactic immunity in the mosquito *Aedes aegypti*. *Sci. Rep* 7, doi:10.1038/srep43163 (2017).

An, S. et al. Insect neuropeptide bursicon homodimers induce innate immune and stress genes during molting by activating the NF-kappaB transcription factor Relish. *PLoS One* 7, e34510 (2012).

Comment 22: Ln 333, ref 28: Please cite the following seminal references here, instead of ref 28.

Moret, Y. and Schmid-Hempel, P., 2000. Survival for immunity: the price of immune system

activation for bumblebee workers. *Science*, 290(5494), pp.1166-1168.

Response: Revised accordingly. We have replaced the previous ref 28 with the recommended citation (Moret & Schmid-Hempel, 2000) (**Line 558**).

Comment 23: Ln 349-350: Add a relevant reference.

Response: Revised accordingly. We have added three relevant references (Moret et al., 2000; Freitak et al., 2003; Bajgar et al., 2015;) to support this statement (**Lines 558**).

Reference:

Moret, Y. & Schmid-Hempel, P. Survival for immunity: the price of immune system activation for bumblebee workers. *Science* (New York, N.Y.) 290, 1166-1168, doi:10.1126/science.290.5494.1166 (2000).

Freitak, D., Ots, I., Vanatoa, A. & Hörak, P. Immune response is energetically costly in white cabbage butterfly pupae. *P Roy Soc B-biol Sci* 270, doi:10.1098/rsbl.2003.0069 (2003).

Bajgar, A. et al. Extracellular adenosine mediates a systemic metabolic switch during immune response. *PLoS Biol.* 13, e1002135, doi:10.1371/journal.pbio.1002135 (2015).

Comment 24: Ln 370-411: This is a long paragraph. It is better to divide it into two paragraphs.

Response: Thank you for the helpful suggestion. We have divided the original long paragraph into two shorter ones to improve readability and logical flow (**Lines 527-556**).

Comment 25: Discussion: What is completely missed in the current manuscript is a discussion of why the NO diet was a better diet for adult immunity than the NP diet. There is a large body of literature reporting nutritional benefits of omnivorous diets in terms of improved protein or nitrogen balance in insects. I think the authors need to discuss about the possibility that improved protein balance on omnivorous diet might have driven insects to invest more protein resources into immunity.

Response: Thank you for raising this important point. In the revised Discussion (**Lines 375-526**), we now explicitly discuss why the NO diet was a better diet for adult immunity than the NP diet, by restructuring the section and incorporating additional mechanistic explanations, literature evidence, and clearer links between improved protein/nitrogen balance during nymphal development and

enhanced immune performance. The discussion is organized around two key components of insect immunity: (1) **constitutive immunity**, defined as baseline immune traits present in the absence of infection, and (2) **inducible immunity**, which is activated upon pathogen challenge, typically amplifying existing constitutive factors and triggering the production of effector molecules such as antimicrobial peptides (AMPs) and lysozyme.

1. Constitutive immunity enhancement potentially driven by higher protein intake during the nymphal stage.

Newly emerged NO adults exhibited higher total protein content compared to NP counterparts. This difference coincided with elevated baseline levels of PPO activity and hemocyte counts under uninfected conditions—hallmark indicators of constitutive immunity—suggesting that early-life omnivory provided greater nutritional resources for immune tissues development and PPO production (see **Lines 375-395** in detail).

2. Early-life omnivory durably enhances adult inducible immune capacity through multiple, non-mutually exclusive pathways:

Despite sharing an identical phytophagous diet during adulthood, NO adults consistently mounted stronger inducible immune responses than their NP counterparts. This enhancement was characterized by higher PPO/PO activity, increased hemocyte proliferation, upregulated immune-related gene expression, and more active transcription of metabolic-related genes associated with energy and biosynthetic support, following *B. bassiana* infection. These effects likely point to at least three non-mutually exclusive mechanisms shaped by early omnivory:

First, omnivorous juveniles accumulated more protein—a critical resource for immune tissue development, PPO production, and AMP synthesis. Notably, this nutritional advantage diminished over time as all adults transitioned to a uniform phytophagous diet, indicating that mechanisms beyond nutrient accumulation contributed to the sustained immune enhancement (see **Lines 375-395** in detail).

Second, early omnivory promoted structural investment into immune systems, as reflected by elevated hemocyte counts at emergence, which likely supported hemocyte proliferation (see **Lines 423-433** in detail) and immune effector production (see **Lines 434-454** in detail) during infection.

Third, early nutritional experience appeared to shape a more robust metabolic infrastructure, capable of meeting the energetic and biosynthetic demands of inducible immunity (see **Lines 473-487** in

detail).

Collectively, these findings suggest that the superior adult immune performance in the NO group may be attributable, at least in part, to improved protein or nitrogen balance during nymphal development.

Comment 26: Figures 3,4,5,6,7, 8: In the same plot, statistically significant differences among treatments were represented by different letters in one case and by asterisks in the other. I think this should be consistent.

Response: Thank you for the suggestion. We have standardized the annotation style across Figures 3-8 in the original submission (now Figures 3, 4, 6, 7, 8, and 9 in the revised manuscript) by using only alphabetic notation to indicate statistical significance. Given that our analyses involved two-way ANOVA, we adopted the following scheme to specify the sources of significance: “X, Y” denote diet effects; “ α , β ” indicate infection effects; and “a, b, c, d” represent significant differences among all treatment combinations when a significant interaction effect was detected. The figure legends have been updated accordingly.

Reviewer #3 (Remarks to the Author):

Comment 1: The manuscript by Li et al. explores the impact of nymphal diet on adult immunity in *Adelphocoris suturalis*. The authors conducted extensive RNAseq experiments to identify differentially expressed immune- and metabolism-related genes in adults fed an omnivorous diet as nymphs compared to those fed a plant-based diet as nymphs. The study demonstrates that an omnivorous diet enhances immune responses and metabolism, leading to higher survival rates following infection with *Beauveria bassiana*. While the study is both interesting and significant, it requires substantial revisions before it can be considered for publication.

Structure, Style, and Clarity

The manuscript would benefit from a thorough revision to improve language, style, and structure. Below are specific suggestions for improvement:

Response: We sincerely thank the reviewer for the careful evaluation of our manuscript and for

providing insightful and constructive comments, which helped improve its clarity, rigor, and interpretability. Following your suggestions, we have revised the Introduction to provide a more detailed background on insect constitutive and inducible immunity, expanded the Methods to enhance transparency in immune gene identification, transcriptome assembly, and gene selection for RT-qPCR validation, and improved figure presentation for clarity and consistency. The Discussion has been substantially expanded to provide deeper mechanistic insights into how early-life diet shapes adult immunity.

Detailed point-by-point responses are provided below.

- Abstract:

Comment 2: The abstract does not clarify the organism or organismal group being studied until the very end. To enhance clarity, I recommend stating at the beginning of the abstract that the study focuses on insect immunity.

Response: Thank you for this helpful suggestion. To improve clarity, we have revised the beginning of the abstract to state that our study focuses on insects. The revised version now opens with:

“In insects, juvenile diets play a pivotal role in shaping adult fitness…” (**Line 11**)

- Introduction:

Comment 3: The introduction would benefit from a more detailed discussion of constitutive and induced immune responses in insects. This is particularly important in the context of the parameters measured in this study (PO activity, PPO activity, and hemocyte load). For instance, how do these parameters in particular contribute to insect immunity, and how are they regulated? Adding this context will help readers understand the significance of the findings.

Response: Thank you for this insightful suggestion. We have expanded the Introduction (**Lines 53-74**) to better explain the distinction between constitutive and induced immune responses in insects, and to clarify how PO activity, PPO levels, and hemocyte load contribute to insect immunity and how they are regulated. Specifically, we elaborated on how pathogen recognition triggers a serine protease cascade that converts PPO into active phenoloxidase (PO), leading to melanization and pathogen sequestration. We also highlight that, upon pathogen recognition, hemocytes proliferate, differentiate, and are mobilized to sites of infection to mediate phagocytosis, nodulation, and

encapsulation. In addition, we clarify that fungal infections activate the Toll pathway via proteolytic cleavage of Spätzle (Spz), leading to downstream signaling through the MyD88-Tube-Pelle cascade and induction of antimicrobial peptides (AMPs) and lysozyme. These revisions provide a clearer mechanistic framework for interpreting the immune traits examined in this study.

- **Figures:**

Comment 4: The figures generally enhance the manuscript's comprehensiveness. However, in Figure 3b, the shown pathway looks very out-of-place and I initially thought it was put there by mistake. Please improve the presentation to clarify that this pathway represents the genes validated by qPCR. This issue is handled much better in Figures 6b, 7b, and 8b.

Response: Thank you for this helpful suggestion. In the revised manuscript, we have redrawn the immune signaling pathway diagram in Figure 3b to improve readability and presentation. The immune-related genes validated by RT-qPCR are now arranged according to their spatial position in the canonical inducible immune signaling cascade—from pathogen recognition and extracellular signal modulation (occurring in the hemolymph) → intracellular signal transduction (signal molecules crossing the membrane into the cytoplasm) → transcriptional activation of immune effector genes (in the nucleus). We believe this arrangement allows readers to more intuitively follow the pathway and better understand the functional roles of each validated immune gene.

Revised Figure 3

Comment 5: For the heatmaps, please explicitly indicate what the colors represent (e.g., fold change) in both the legend and the figure captions.

Response: Thank you for the helpful suggestion. We have now revised the figure captions and legends to clearly indicate what the colors in the heatmaps represent. Specifically, the color scale represents Z-scores computed by standardizing the expression of each gene across all samples. Red indicates expression higher than the average level of that gene across samples ($Z > 0$), blue indicates lower expression ($Z < 0$), and white indicates near-average levels ($Z \approx 0$). These modifications have been implemented in all relevant figures (Figures 3a, 7a, 8a, and 9a in the revised manuscript)

and their captions/legends to enhance clarity and interpretability.

Methods

Comment 6: The methods are described in good detail overall, but some sections lack critical information and clarity. Below are specific areas that need attention:

- Identification of Immune-Related Genes:

- o The authors mention using tBLASTn searches to identify immune-related genes (Line 161). However, the description is incomplete. Please provide details on the query and database used, the search parameters, and how the immune-related genes were identified.

Response: Thank you for your thoughtful suggestion. We have revised the Methods section to provide a more detailed and transparent description of the immune gene identification procedure. Specifically, “a reference set of insect immune-related genes was compiled from NCBI, incorporating well-annotated protein sequences from both model organisms and hemipteran pests, including *Drosophila melanogaster*, *Anopheles gambiae*, *Apis mellifera*, *Bombyx mori*, *Tribolium castaneum*, *Nilaparvata lugens*, *Aphis gossypii*, and *Halyomorpha halys*. These sequences were used as queries in a tBLASTn search against the *A. suturalis* unigene dataset using Basic Local Alignment Search Tool v2.16.0, with an E-value cutoff of $1e^{-5}$. The resulting hits were manually curated and further validated by conserved domain architecture using CD-search against the NCBI Conserved Domain Database (CDD). Only unigenes containing domain structures consistent with known immune-related gene families were retained.” These additions have been incorporated into the revised Methods section (**Lines 712-723**).

Comment 7: Selection of Genes for RT-qPCR Validation:

- o The manuscript briefly mentions the validation of immune-related genes (Lines 179–182) and macro-nutrient metabolism pathway genes (Lines 256–271) via RT-qPCR. Please elaborate on how these genes were selected and the rationale behind their selection.

Response: We appreciate the reviewer's comment regarding the selection of genes for RT-qPCR validation. We have clarified this selection strategy in the revised Methods (**Lines 724-737**) to improve transparency and reproducibility. Specifically, the approach is as follows: Candidate genes used for RT-qPCR validation were selected from the identified differentially expressed gene (DEG)

sets based on three criteria: pathway relevance, statistical significance, and functional annotation. For immune-related genes, we focused on key components of inducible antifungal immune pathways that were significantly enriched in our transcriptomic analysis, including fungal recognition factors, core components of the Toll signaling pathway, and antifungal effectors. For nutrient metabolism-related genes, we selected central nodes in anabolic or catabolic pathways that are functionally linked with immunity. When multiple members of a gene family met these criteria, we prioritized those showing the largest differential expression between NO-Bb and NP-Bb groups, higher sequence conservation, complete functional domains, and well-supported functional annotation as representative genes. We consider that this approach ensures the validated genes are representative and biologically meaningful.

Comment 8: Transcriptome Assembly Quality:

o The authors de-novo assembled a transcriptome, but I cannot find anything about the quality of the assembly. In the methods, the authors state that they used BUSCO (Line 491: for retrieving transcript ORF? Please clarify this). Please provide information on the software version, reference database, and quality metrics (e.g., completeness scores). Additionally, software versions should be included throughout the methods section for consistency.

Response: Thank you for the constructive comment. We have revised the Methods section to provide a complete description of the transcriptome assembly and quality assessment. Specifically, we now report key assembly statistics (total number of transcripts, N50, mean transcript length), and we clarify that BUSCO v3.0.2 was used solely to assess assembly completeness using the insecta_odb10 database, which showed 96.7% complete orthologs. The earlier inaccurate statement regarding BUSCO's function has been corrected. We have also added the specific software versions in the Methods section: Fastp v0.19.5, Trinity v2.8.5, TransRate v1.0.3, CD-HIT v4.5.7, DIAMOND v0.9.24, and HMMER 3.2.1 (**Lines 691-708**)

Discussion

Comment 9: The discussion would benefit from a deeper exploration of the mechanisms underlying the observed results. Specifically:

- The authors should discuss how the upregulation of immune-related genes in the omnivorous diet

(NO) group enhances survival following *B. bassiana* infection. More information on the infection route of *B. bassiana* and the roles of PO, PPO, and hemocytes in combating its spores would strengthen both the introduction and the discussion. This would provide important context for understanding the observed immune responses.

Response: We appreciate this valuable suggestion. We have extensively expanded both the Introduction and Discussion to provide deeper mechanistic insight while adding the pathogen-specific context requested.

1. Added pathogen-specific background

- **Infection route of *B. bassiana*:** Conidia adhere to the insect cuticle, penetrate within 24-48 h, invade the hemocoel, and trigger systemic immune activation (Introduction, **Lines 87-89**; Discussion, **Lines 403-407**).
- **Roles of PO, PPO, and hemocytes in antifungal defense:** PPO/PO-mediated melanization generates reactive quinone and melanin to encapsulate and neutralize fungal hyphae, preventing further tissue colonization. In parallel, hemocytes aggregate at infection sites, and together with melanization, drive nodule formation that restricts fungal spread and facilitates pathogen clearance. these responses effectively confine and eliminate invading fungi (Introduction, **Lines 61-70**; Discussion, **Lines 462-470**).

2. Linked immune gene upregulation to enhanced antifungal defense

We highlight that, compared with NP-Bb adults, the immune-related genes upregulated in NO-Bb adults were predominantly enriched in fungal recognition, the fungal-responsive Toll signaling pathway and its downstream antifungal effectors (e.g., Alo, lysozymes), likely enhancing fungal clearance and explaining their higher post-infection survival (Discussion, **Lines 458-462**)

3. Expanded the mechanistic discussion of early-life diet-mediated immune modulation

We have restructured the Discussion to analyze how early-life omnivory enhanced adult immunity from two complementary perspectives:

- **Constitutive immunity** (Discussion, **Lines 375-395** in detail): Newly emerged NO adults exhibited higher total protein content compared to NP counterparts. This difference coincided with elevated baseline levels of PPO activity and hemocyte counts under uninfected conditions, suggesting that early-life omnivory provided greater nutritional resources for immune tissues development and PPO production.

- **Inducible immunity** (Discussion, **Lines 396–487** in detail): Despite identical adult diets, NO adults showed higher PPO/PO activity, hemocyte counts, immune-related gene expression, and metabolic-related gene transcription after *B. bassiana* infection. These effects likely point to at least three non-mutually exclusive mechanisms shaped by early omnivory: 1) Omnivorous juveniles accumulated more protein—a critical resource for immune tissue development, PPO production, and AMP synthesis (**Lines 408–422** in detail). 2) Early omnivory promoted structural investment into immune systems, as reflected by elevated hemocyte counts at emergence, which likely supported hemocyte proliferation (**Lines 423-433** in detail) and immune effector production (**Lines 434-454** in detail) during infection. 3) Early nutritional experience appeared to shape a more robust metabolic infrastructure, capable of meeting the energetic and biosynthetic demands of inducible immunity (**Lines 473–508** in detail).

Minor Comments

Comment 10: Line 50: Please explain the abbreviation "P:C."

Response: Thank you for pointing this out. "P:C" refers to the protein-to-carbohydrate ratio. We have now spelled this out in full in the revised manuscript to improve clarity (**Line 46**).

Comment 11: Line 141: Clarify what is meant by "-log10 values."

Response: Thank you for the suggestion. We have clarified in the text that "–log10 values" refer to "–log10 [adjusted p-value]", which used to represent statistical significance in the volcano plots (**Line 150**).

Comment 12: Figure 2: The font size is too small and should be increased for better readability.

Response: We appreciate this helpful comment. The font size in Figure 2 has been increased to improve readability (see Revised Figure 2).

Comment 13: Figure 2c, e: The color codes in panels c and e are inconsistent. Please standardize the color codes for easier comparison. Additionally, using the same x-axis scale for both plots would improve visual comparability.

Response: Thank you for the constructive comment. We have standardized the color codes and

unified the x-axis scales in Figure 2c and 2e to improve consistency and facilitate comparison (see Revised Figure 2c, e).

Response to Reviewers

We sincerely thank the reviewers for their thoughtful and constructive comments, which have helped us improve the clarity, conciseness, and rigor of our manuscript. All changes in the revised manuscript are highlighted in yellow for ease of review. Below, we provide detailed point-by-point responses.

Reviewer #1 (Remarks to the Author):

Comment 1: Thank you very much for the opportunity to review the revised version of the manuscript titled “Nymphal diets boost adults’ immunity via strengthened constitutive immunity and metabolic capacity in a mirid bug.” The authors have addressed all of my suggestions, mostly in a very satisfactory manner, including the performance of new experiments. The additional information added to the main text has led to better clarification of the methodology used and improved data presentation.

I also appreciate that the authors thoroughly revised the Discussion section and provided a more specific explanation of the observed physiological changes in the mirid bug following different dietary treatments. However, in my opinion, the current version of the Discussion is too long, as the authors have added approximately nine pages of new information and explanations. I believe this section could be shortened, and the authors should focus on extracting and presenting the most important points.

Response: We thank the reviewer for this constructive suggestion, which helped us improve the clarity and readability of the Discussion. We agree that the previous version was unnecessarily long. In the revised manuscript, we shortened the Discussion by approximately 31% in word count (from ~2,568 to ~1,763 words) and reduced it from 16 to 13 paragraphs, focusing on the most important findings.

Major changes include:

- 1. Integrated metabolism discussion:** The original sections on carbon/lipid metabolism (original P8), amino acid metabolism (original P9), and the strong immunity-low consumption phenomenon (original P13) were merged into a single integrated paragraph

(now P8). Overly detailed background descriptions were condensed, with only the core functional information retained.

2. Consolidated hypotheses: The two separate hypothesis paragraphs (original P11–12) were condensed into one concise discussion (now P10).

3. Deleted/shortened redundant content: The digression on lower triglyceride levels in NO adults (original P4) was removed, and the extended discussion of the strong immunity-low consumption phenomenon (original P13) was compressed to its key observations (now P8).

4. Improvements in language and presentation: Redundant phrasing was condensed into more concise formulations, producing shorter and clearer sentences.

These revisions have made the Discussion substantially clearer, more concise, and easier to follow, while preserving the depth and rigor of the analysis.

Reviewer #3 (Remarks to the Author):

Comment 1: I appreciate the authors' detailed responses and think they significantly revised and improved the manuscript. However, I still have a few minor issues that I recommend addressing.

Response: We thank Reviewer #3 for the positive evaluation of our previous revision. We also appreciate the additional minor suggestions and have carefully addressed them in the revised manuscript. Detailed responses follow below.

Comment 2: Comment 6:

The authors have added a more detailed description of the immune gene identification in the Methods section, but the general methodology should also be apparent from simply reading the results. However, in Line 170, it only says “Using tBLASTn searches, we identified 695 immune-related unigenes”, which does not provide enough clarity for readers. Please rephrase this as follows: “Blasting a reference set of insect immune-related genes against the *A. suturalis* genome, we identified ...”

Response: Revised as suggested. The sentence now reads: "*Blasting a reference set of insect immune-related genes against the *A. suturalis* transcriptome, we identified 695 immune-related unigenes.*" (Lines 170-171)

Comment 3: Comment 8:

The authors have clarified the use of BUSCO and added the missing information about software versions, reference database and some quality metrics. However, I would appreciate if they gave the complete BUSCO result, including duplicated BUSCOs, since this is important to evaluate the quality of the transcriptome.

Response: As suggested, we now provide the complete BUSCO results, including duplicated BUSCOs. The sentence now reads: "*Assembly completeness was assessed using BUSCO v3.0.2 with the insecta_odb10 database, which showed 96.7% complete (57.6% Single-copy; 39.1% duplicate).*" (Lines 616-617)

Additional minor points:

Comment 4: Please make sure to use the terms “nymph” and “larva” correctly for hemimetabolous and holometabolous insects, respectively (line 46, for example).

Response: Corrected throughout. We now consistently use “nymph” for hemimetabolous insects and “larva” for holometabolous insects. (Lines 38, Line 46, Line 52)

Comment 5: Please change the term “nymph diet” to “nymphal diet” throughout the manuscript

Response: Revised as suggested. We have replaced “nymph diet” with “nymphal diet” throughout the manuscript to ensure consistency. (Line 132, Line 321)

Comment 6: Line 76-77: rephrase: “... that can cause serious damage to various crops including cotton, soybean, maize and fruit trees”

Response: Thank you. We have revised the sentence as suggested to: “*Adelphocoris suturalis*

(Hemiptera: Miridae) is a notorious pest that can cause serious damage to various crops including cotton, soybean, maize and fruit trees." (Lines 75-77)